# X-ray parametric down-conversion reveals EUV-polariton

Dietrich Krebs [1,2,3] ✉, Fridtjof Kerker[2,3], Xenia Brockmüller[3], Christoph J. Sahle [4], Blanka Detlefs [4], Simo Huotari [5], Nina Rohringer [1,2,3,6] & Christina Bömer [1,2] ✉

Spontaneous parametric down-conversion (PDC) of photons is a gateway into the quantum realm – thoroughly studied in nonlinear optics and ubiquitously used to generate non-classical states of light. Extending PDC from the visible regime towards shorter wavelengths further enables microscopic resolution of electronic structure and quantum-enhanced X-ray detection, but remained challenging due to the process' inherently low conversion rate. Here, we resolve the full signal cone of non-degenerate down-conversion at X-ray wavelengths and identify imprints of a polariton in the extreme ultraviolet (EUV) regime. We confirm our finding of the EUV-polariton with theoretical simulations and establish that our approach directly images the characteristic anti-crossing of polaritonic dispersion branches. This insight could open a pathway to explore strong-coupling phenomena of EUV-light-matter interaction.

Polaritons are light-matter hybrid states that emerge when photons couple strongly to excitations within matter. First identified in phonon–photon[1] and electron–photon coupling[2,3], polaritons have since been confirmed for a wide variety of systems[4] and proved instrumental in tailoring quantum materials[5–7]. By use of resonant cavities, material properties from conductivity and magnetism[8–10] all the way to chemical behavior[11–15] and superconductivity[16,17] can be altered upon reaching the strong-coupling regime.

These impressive advances in cavity-quantum electrodynamics (QED), using visible and near-infrared light, are mirrored in circuit-QED for radio-frequency strong-coupling[18–20]. At shorter wavelength, analogous polaritonic hybridization stands to be expected, given the universal nature of QED. However, the lack of suitably reflecting cavities has so far precluded the realization of strong-coupling conditions under the conventional, cavity-based paradigm—leaving polaritonic phenomena largely unexplored in the EUV and X-ray range.

In this work, we present evidence for polariton formation in the EUV via a different route and thus open up this regime of light-matter hybridization for investigation. Our approach exploits non-degenerate

XPDC to both excite and probe an EUV-polariton, which derives from one of the down-converted photons, while imprinting its signature onto the other, correlated photon.

In order to access XPDC efficiently, we introduce a momentum-resolved detection scheme, combining the imaging capabilities of a bent crystal analyzer with 2-dimensional acquisition on a pixel detector. This enables us to resolve the signal cone of non-degenerate XPDC for the first time and reveal its modulation by the EUV-polariton. Observing its excellent agreement with theoretical simulations, we obtain further indication of EUV-strong-coupling and conclude significant potential for future explorations.

## Results

### PDC in the X-ray regime

XPDC can be described as a nonlinear diffraction process, in which an incident "pump" photon—denoted by its wavevector $\mathbf{k}_p$ – scatters off a single crystal sample and thereby splits into a strongly correlated pair of photons $\mathbf{k}_s$ and $\mathbf{k}_i$ (Fig. 1a). These are conventionally referred to as 'signal' and 'idler', respectively[21]. Similar to the visible regime[22], the

[1]Deutsches Elektronen-Synchrotron DESY, Hamburg, Germany. [2]The Hamburg Centre for Ultrafast Imaging, Hamburg, Germany. [3]Department of Physics, Universität Hamburg, Hamburg, Germany. [4]ESRF, The European Synchrotron, Grenoble, France. [5]Department of Physics, University of Helsinki, Helsinki, Finland. [6]Center for Free-Electron Laser Science (CFEL), Deutsches Elektronen-Synchrotron DESY, Hamburg, Germany. ✉e-mail: dietrich.krebs@desy.de; christina.boemer@desy.de

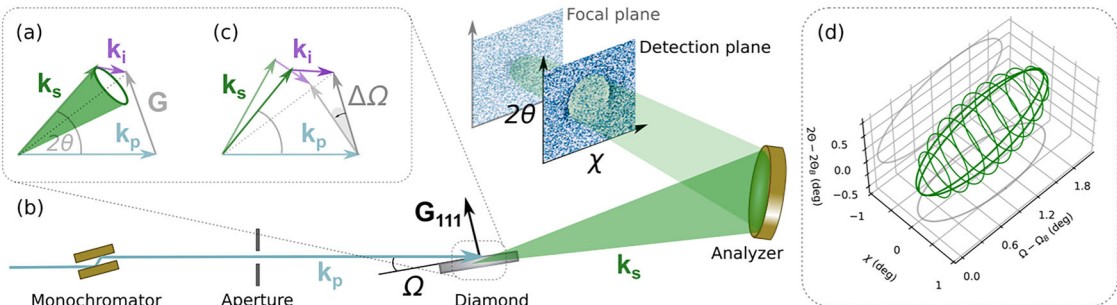

**Fig. 1 | Principles of XPDC and its detection. a** XPDC is phase-matched in diffraction geometry, using the reciprocal lattice vector $G_{111}$ of diamond to satisfy momentum conservation $\mathbf{k}_p + G_{111} = \mathbf{k}_s + \mathbf{k}_i$. The signal ($\mathbf{k}_s$) is emitted on a rotationally symmetric cone. **b** The experiment (schematic) uses a pump beam monochromatized at 9.79 keV (light blue), which is down-converted into 9.69 keV signal photons (green) plus unobserved idlers (100 eV, violet). The signal cone (shaded in green to indicated the 3D extend) is filtered and re-focused by a spherically-bent analyzer before it is imaged onto a 2D-pixel detector outside the focal plane. **c** Depending on the rotation of the sample ($\Delta\Omega$), a range of phase-matching solutions is admissible, resulting in signal cones of varying radii. **d** These solutions form an ellipsoid in scattering space (green), bordering the Bragg spot at ($\Omega_B$, $2\theta_B$, $\chi = 0$) and extending towards larger sample angles $\Omega$.

down-conversion process will only occur if its phase-matching condition is satisfied, which is fundamentally determined by energy and momentum conservation. For X-rays, this involves a reciprocal lattice vector (**G**) of the sample, governing the direction of nonlinear diffraction: $\mathbf{k}_p + G_{111} = \mathbf{k}_s + \mathbf{k}_i$. Around the central direction, $\mathbf{k}_s$ and $\mathbf{k}_i$ are emitted into scattering cones, for which the phase-matching condition is rotationally symmetric (cf. Fig. 1a). These cones constitute the characteristic scattering signature of (X-ray) PDC[23]. While the cone-shaped emission is well known in the visible range (e.g., ref. [24]), it could only recently be observed for X-rays in the degenerate[25] and—here—the non-degenerate case of XPDC.

In our experimental realization of XPDC at the European Synchrotron Radiation Facility, we adapt the large-solid-angle spectrometer of beamline ID20[26] to resolve the signal cone spatially and spectrally (Fig. 1 b). We employ a monochromatized pump beam at $\hbar\omega_p = 9.79$ keV photon energy (~$10^{13}$ photons/s) to scatter off a diamond sample (1 mm thick; (111)-surface-cut), which is hit under an angle $\Omega$. Regular Bragg diffraction occurs for $\Omega_B = 17.91°$, while larger rocking angles allow phase-matching for nonlinear diffraction, i.e., XPDC. We focus specifically on non-degenerate (i.e., asymmetric) down-conversion into signal photons at $\hbar\omega_p = 9.69$ keV and idler photons at $\hbar\omega_i = 100$ eV (fixed by energy conservation $\omega_p = \omega_s + \omega_i$). We filter specifically for the signal photon using one of the spectrometer's spherically-bent crystal analyzers (SBCA), while the idler remains unobserved, due to its interaction with the material.

This SBCA (1 m bending-radius; Si(660)-surface) forms the centerpiece of our detection scheme: Operating it close to backscattering geometry, on its Rowland circle[27], we can capitalize on the analyzer's imaging capabilities (cf. ref. [28]) to fully resolve the signal cone. As the SBCA reflects an energetically filtered mirror image of the original scattering distribution (cf. Fig. 1b), we find the signal cone being re-focused with all relative angles preserved. Inserting a 2D pixel-detector (MAXIPIX[29]) before the analyzer's focal plane, we intersect and image the re-converging cone. Finally, we map from pixels to scattering angles (Methods 2), denoting the in-plane angle as $2\theta$ (cf. Fig. 1a) and out-of-plane angle as $\chi$. We can equivalently access the momentum transfer $\mathbf{q} = \mathbf{k}_p - \mathbf{k}_s$ of the scattering process (Methods 3) and, hence, refer to our detection scheme as 'momentum-resolved'.

The introduction of 2D detection in combination with an imaging analyzer distinguishes our setup fundamentally from all previous approaches to measure XPDC (e.g., refs. [23,30–36]). It improves our efficiency in data-acquisition substantially, as we can capture the entire XPDC signal cone in a single frame, rather than resolving it by point-wise raster scan (cf., e.g., ref. [23]). This, effectively, enables us to map out the phase-matching condition.

## Catching the cone

We expect the signal cone to vary in size across the range of phase-matching angles ($\Omega$). Rotating the sample by $\Delta\Omega = \Omega - \Omega_B$ from the Bragg condition, we tune the nonlinear diffraction geometry, tilting the reciprocal lattice vector $G$ (Fig. 1 c). The signal cone's opening angle follows these adjustments, mapping out an ellipsoid in scattering coordinates ($2\theta$, $\chi$) versus $\Omega$ (Fig. 1d). This constitutes the characteristic phase-matching signature of XPDC[23]. Confirming this pattern will allow us to unambiguously identify XPDC without coincident detection of any idler photon.

Measuring individual slices through the ellipsoid (at fixed $\Delta\Omega$), we can trace phase-matching from $\Delta\Omega = 0.27°$ up to 2.07°. In Fig. 2a–c, we present three examples of the imaged XPDC cones at: $\Delta\Omega_a = 0.47°$, $\Delta\Omega_b = 1.07°$ and $\Delta\Omega_c = 1.87°$, respectively (positions relative to $\Omega_B = 17.91°$). All images are background subtracted, as the weak XPDC signature was otherwise obscured by Compton scattering (occurring at ~3 counts/(pixel*s)). Using suitably scaled background images (Methods 4), its smooth contribution can be removed, revealing the underlying XPDC signal at up to 0.4 counts/(pixel*s). While this magnitude is consistent with our theoretical expectations[37], the distribution of the signal is surprising: We discover *two* conjoined scattering cones, rather than a single one. Primarily, we find a positive, circular pattern inside each phase-matching cone, where the scattering yield peaks above the background level (bright color in Fig. 2a–c). In addition, we observe a negative feature, lining the perimeter of the first cone (dark color). Here, count rates fall consistently below the average background level.

This two-fold modulation becomes even more apparent if visualized on a divergent color-scale (Fig. 2d). In all slices, the change of sign aligns with the phase-matching ellipsoid of Fig. 1d (reproduced as a green wire-frame in Fig. 2d). Fitting the circular contrast at this change of sign, we extract the cones' opening angles and confirm their agreement with phase-matching predictions quantitatively: For Fig. 2 a–c, this respectively yields 0.97° (vs. 0.97°), 1.20° (vs. 1.17°) and 0.62° (vs. 0.57°)—predictions juxtaposed in brackets. Across all acquisitions (cf. Methods 1) the average deviation from the phase-matching signature[23] amounts to 4.5%, ascertaining our observation as XPDC. The close agreement, furthermore, suggests an intimate connection of the two-fold modulation with XPDC, even though the pattern stands in stark contrast to the prototypical emission characteristic of visible PDC (i.e., uniformly positive signal cones[24]). To explain this anomaly, we introduce a polaritonic model of XPDC, identifying the two-fold pattern with two distinct dispersion branches of an EUV-polariton.

Before discussing its details, we briefly address a second observation: All XPDC cones of Fig. 2 exhibit systematically reduced

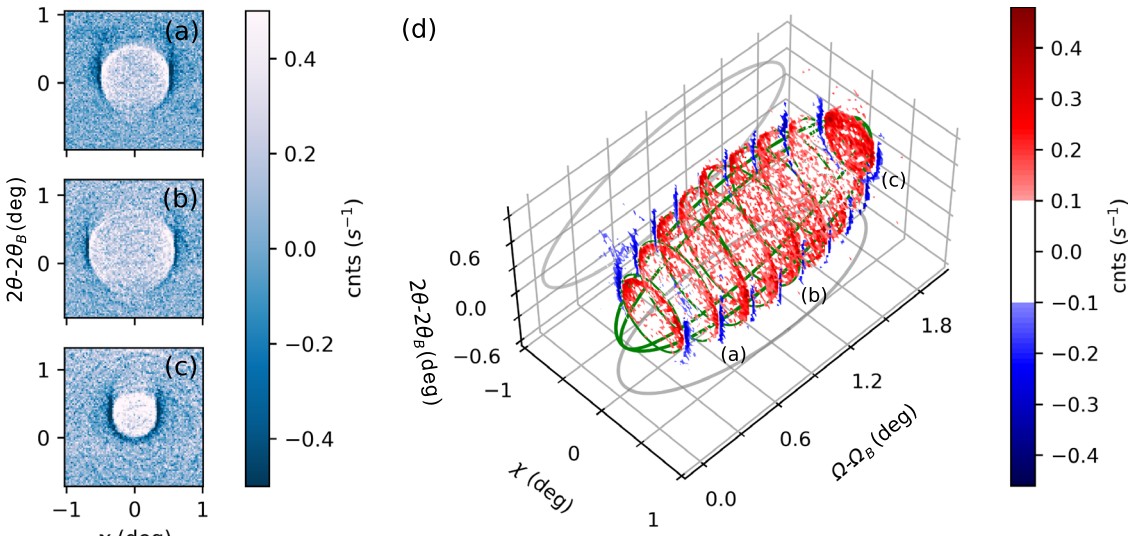

**Fig. 2 | Measurements of the XPDC cone. a–c** Circular scattering patterns recorded for phase-matching at $\Delta\Omega_a = 0.47°$, $\Delta\Omega_b = 1.07°$ and $\Delta\Omega_c = 1.87°$ above the elastic (111)-reflection in diamond, respectively. All data is background subtracted and plotted on a common color scale. Cone openings amount to 0.97°, 1.20° and 0.62° in diameter, respectively. **d** Enhanced contrast (divergent color scale) visualizes the excellent agreement of measured XPDC with its theoretically predicted signature (green, ellipsoidal wire-frame) across the complete phase-matching region ($\Delta\Omega = [0.27, \ldots 1.87°]$). Projections of the ellipsoid are shown in gray. The striking two-fold modulation of the scattering pattern can be related to different polaritonic contributions to XPDC (red/blue: predominantly upper branch/lower branch).

scattering signals at small values of $\chi$. The negative circles (dark/blue) show particularly pronounced openings across the vertical. We previously predicted this effect for XPDC involving optical photons[23,37], linking it to polarization constraints on the idler (cf. Methods 5). Our detection scheme allows us to experimentally resolve this effect for the first time.

## A polaritonic picture of XPDC

Polaritons emerge as light-matter hybrid states, when bare photons couple strongly to material excitations[1,2,4]. In the present case, this applies to the idler photon: Emerging at $\hbar\omega_i = 100$ eV $= \hbar c(|\mathbf{k}_p| - |\mathbf{k}_s|)$ from the scattering process, it couples to highly-excited electronic states within the diamond sample. There, repeated absorption and subsequent re-emission of the original idler can effectively dress it into a polariton (see: cyclic illustration in Fig. 3a). Upon reaching the sample's surface, the propagating polariton decays into either of its constituents, i.e., the bare idler photon ($\gamma$) or an excited, free electron ($e$). Depending on this final state, we may consider the overall process as parametric conversion ($\mathbf{k}_p \to \mathbf{k}_s + \mathbf{k}_i$) or Compton scattering ($\mathbf{k}_p \to \mathbf{k}_p + e$), respectively. Reconciling both processes in a common precursor is a particular strength of the polaritonic interpretation and will help to address the long-standing conundrum about their hypothetical interference[30,35].

To transfer our polaritonic interpretation of XPDC into an effective scattering description, we start from a two-level system (TLS) as our model. This approach corresponds to the low-excitation limit of a collectively-coupled system of emitters[14,38,39] and captures the effective polaritonic response of the material (cf. Methods 6). We choose the basis states of the TLS to correspond to the polariton's photonic $|\phi_\gamma\rangle = (1,0)^T$ and electronic $|\phi_e\rangle = (0,1)^T$ components and write the associated Hamiltonian as

$$\hat{H}^{POL} = \hbar \begin{pmatrix} \omega_\gamma & 0 \\ 0 & \omega_e \end{pmatrix} + \begin{pmatrix} 0 & V \\ V^* & 0 \end{pmatrix}. \quad (1)$$

Here, the diagonal entries account for the bare states' energies, while the off-diagonals introduce an effective light-matter coupling $V$, which causes the polaritonic hybridization. Diagonalizing $\hat{H}^{POL}$, yields the well-known eigen-energies of a coupled TLS

$$E_\pm = \left( \frac{\hbar(\omega_\gamma + \omega_e)}{2} \pm \sqrt{\frac{\hbar^2}{4}(\omega_\gamma - \omega_e)^2 + |V|^2} \right) \quad (2)$$

for the corresponding dressed states $|\phi_\pm\rangle$. Upon dispersion in $\omega_\gamma$, the energies trace out the upper (+) and lower (−) polariton branches, illustrated in Fig. 3b. Capturing the characteristic anti-crossing around $\omega_\gamma = \omega_e$, the model's dispersion branches display the essential phenomenology of polaritonic hybridization[1,2,4].

Extending Eq. (2) across the scattering plane for XPDC (see Fig. 3c), we relate its angles $2\theta$, $\chi$ to momentum transfer (Methods 3) before connecting the bare TLS energies to the scattering kinematics: Energy transfer determines the effective electronic excitation as $\omega_e \sim \omega_p - \omega_s$, where energy conservation is relaxed due to the short-lived nature of the excited state. Conversely, momentum transfer determines the photonic level via the vacuum dispersion relation

$$\omega_\gamma(\mathbf{q}) = c|\mathbf{k}^{eff}_\gamma| = c|\mathbf{q} + \mathbf{G}| = c|\mathbf{k}_p - \mathbf{k}_s + \mathbf{G}|. \quad (3)$$

As the result of this mapping, we find the polaritonic branches $E_{+/-}$ transformed into dispersion surfaces, that align with a hyperboloid of rotation (shown in Fig. 3 c for fixed $\Delta\Omega = 1.07°$ and $V = 0.82$ eV). The avoided crossing of branches becomes a circular seam, which coincides with the phase-matching condition predicted for 'regular' XPDC (green dashed circle). Tracing this correspondence on to the scattering pattern—juxtaposed as the bottom layer of Fig. 3c—we observe that the two-fold modulation coincides with the anti-crossing seam as well. These matching signatures reinforce our polaritonic interpretation of XPDC, anticipating the direct mapping shown below.

## From weak scattering to strong coupling

For direct comparison of the polaritonic model with our measurements (Fig. 1), we embed the TLS into a simulation of inelastic X-ray scattering. Modifying the dynamic structure factor S($\mathbf{q},\omega$)[40] to incorporate the polariton's Hopfield coefficients[2], we obtain the required scattering cross section or yield per pixel (Methods 7, Eqs. (13, 14)). In Fig. 4a, b, we compare half of a measured XPDC cone to our respective

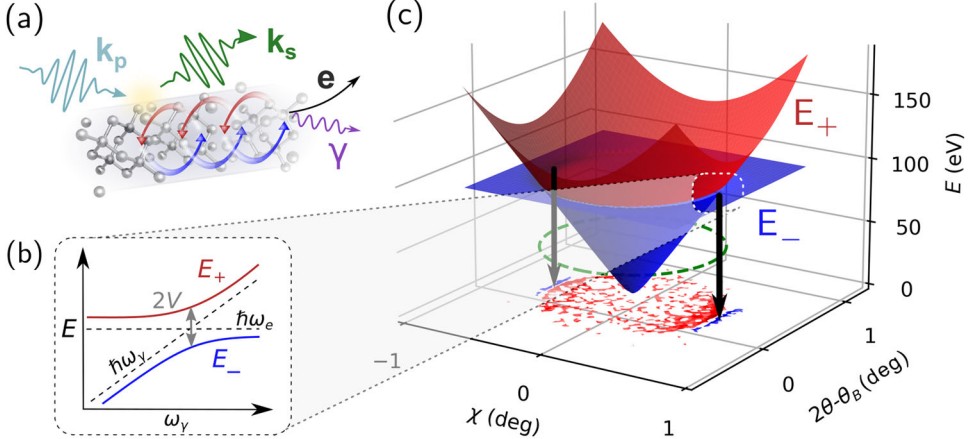

**Fig. 3 | Illustrations of EUV-polariton characteristics. a** Down-conversion from pump (**k**_p, blue pulse) to signal (**k**_s, green pulse) photon transfers -100 eV = $\hbar c(|\mathbf{k}_p|-|\mathbf{k}_s|)$ into diamond, launching an EUV-polariton. Repeated absorption of the energy into electronic excitations and re-emission as a photon hybridize both states (cyclical coupling depicted in red-blue). The resulting EUV-polariton, ultimately, decays into either of its constituents, a photon (γ, violet) or a free electron (**e**, black) −matching the kinematics of XPDC or Compton scattering, respectively. **b** Polaritonic dispersion relation shown schematically for upper (E_+) and lower (E_−)

polariton branch; dotted lines indicate the uncoupled photonic ($\omega_\gamma$) and electronic ($\omega_e$) excitations. Their hybridization opens an avoided crossing (Rabi splitting) of 2 V around $\omega_e = \omega_\gamma$. **c** Polariton dispersion mapped across the scattering plane (2θ, χ) of XPDC; the previous branches E_{+/−} (cf. inset **b**) transform into dispersion surfaces that anti-cross at the phase-matching condition (green, dashed circle). The seam mirrors the two-fold modulation observed in the scattering pattern (bottom layer)−both cases show $\Delta\Omega = 1.07°$.

TLS-based simulation, choosing a slice of the phase-matching ellipsoid at $\Delta\Omega = 1.07°$ (cf. Fig. 1 a) and parameters as extracted below. We find excellent visual agreement, with the opening angle being correctly reproduced at 1.20° and the azimuthal polarization-imprint yielding the expected minima around χ = 0° (Methods 5). Most importantly, our model accurately captures the two-fold modulation of polaritonic XPDC around the cone's perimeter.

The modulation's magnitude is determined by the coupling strength V, which remained a free parameter of the Hamiltonian (Eq. 1) so far. To extract its physical value, we fit a central line-out of the scattering cone at $2\theta - 2\theta_B = 0.25°$ (Fig. 4 c, green data) with our TLS-based model (Methods 8). Resulting in $V = 0.82 \pm 0.08$ eV and an overall broadening $\hbar\Gamma = 1.64 \pm 0.22$ eV, the best fit is given by the blue−red curve. Its color-coding reveals further insight into scattering contributions: Marking the model-signal blue, where it stems predominantly from the lower polariton branch (E_−), versus red for the upper branch (E_+), we can directly identify the negative and positive modulations of the scattering pattern with the respective polariton branch. This confirms our hypothesis that the polaritonic dispersion is, indeed, mapped by XPDC (cf. Fig. 3c), which opens a window to explore light-matter hybridization in the EUV. Having thus obtained a sensitive probe of the anti-crossing region (Fig. 4 d), we focus again on the level splitting found there (zoomed inset) and put its magnitude into perspective: At $2V = 1.64$ eV, the Rabi splitting is comparable to the overall broadening $\hbar\Gamma = 1.64$ eV. This condition $(2V \gtrsim \hbar\Gamma)$−alongside the visibility of the splitting itself−is conventionally considered to mark the onset of the strong-coupling regime[5,6,12,14,15,39]. As such, our results indicate that the EUV-polariton could reach strong-coupling conditions intrinsically. This is particularly remarkable in contrast to more typical strong-coupling schemes, which require cavity-enhancement[4–7], and thus encourages to further explore the underlying coupling mechanism.

## Discussion

We reported on the first observation of a full signal cone for non-degenerate PDC in the X-ray regime, employing a momentum-resolved detection scheme. The measured scattering patterns reveal unexpected modulations, indicating that a polaritonic substructure is imprinted onto XPDC. Expanding upon this interpretation, we develop

a polaritonic TLS-model, confirm its viability and conclude the existence of an EUV-polariton. This light-matter hybrid state further exhibits hallmarks of strong coupling—most notably in the absence of any resonant cavity.

By extending the strong-coupling paradigm into the EUV spectral range, several enticing research avenues unfold: Understanding light-dressed states in condensed matter will facilitate the transfer of EUV-spectroscopy techniques from the gas phase, thus advancing applications at high-harmonic[41] and free-electron laser[42] sources. Moreover, the EUV-polariton's intrinsically short wavelength (-10 nm) recommends it as a probe of nanoscopic structures, sensed while traversing a sample. Embedding such diagnostic function with EUV-lithography[43,44], the control of light-matter hybridization appears as a long-term perspective. Meanwhile, the unique access provided to the propagating EUV-polariton by XPDC may elucidate fundamental aspects of bulk polaritons and the role of cooperativity in collective strong-coupling phenomena[14,45–47].

Concluding on the original merit of (X)PDC, namely, the production of entangled photons, we observe the enticing possibility to combine X-ray quantum optics with polaritonic hybridization: By extending entanglement from photon pairs partially onto the EUV-polariton, non-classical states of light and matter could be envisioned.

## Methods

### Experimental setup

We performed the experiment at the large solid-angle spectrometer of beamline ID20 at ESRF[26]. The setup's geometry follows Fig. 1c. The incident beam passes through a Si(111)-double-crystal monochromator, limiting its spectral bandwidth to -1 eV (FWHM) around the pump photon energy of $\hbar\omega_p = 9.79$ keV at a fluence of $-10^{13}$ ph/s. As our sample, we use a diamond single-crystal of 1 mm thickness and (111) surface-cut. This is initially positioned for its symmetric (111) Bragg-reflection at $\Omega = \theta_B = 17.91°$ (nominal calibration) and subsequently aligned to fulfill phase-matching conditions at $\Omega = \theta_B + \Delta\Omega$ (measurements for $\Delta\Omega = [0.27,...., 1.87°]$, see Fig. 5 below). We filter for the XPDC signal using a Si(660) crystal analyzer, which is spherically bent with a 1 m curvature radius. It is positioned at a (vertical) scattering angle of $2\theta_B$ in - 1 m distance downstream of the

sample (viz., the so-called Rowland geometry). The analyzer is aligned close to back-reflection ($\theta_{analyzer}$).

## Angle calibration

We calibrate the 2-dimensional detector images from pixels to scattering angles [deg] by means of the sample's (111) Bragg-reflection. This locates the (nominal) scattering angle $2\theta = 2\theta_B$, corresponding to pixel 117 vertically, and also defines the scattering plane around $\chi = 0$, corresponding to pixel 79 horizontally. The scale relative to this reference is obtained by moving the detection-arm (i.e., the assembly including analyzer and detector) along $2\theta$ by $\pm2°$. We convert the pixel positions that the Bragg reflex traverses on the detector during this movement into the calibration: 1 pixel $\triangleq$ 0.0205°.

This holds for both the $\chi$- and the $2\theta$-axes, as the pixels are quadratic.

$$2\theta[\text{deg}] = (2\theta\,[\text{pixel}] - 117\,\text{pixel}) \times 0.0205\,\text{deg/pixel} \quad (4)$$

$$\chi[\text{deg}] = (\chi\,[\text{pixel}] - 79\,\text{pixel}) \times 0.0205\,\text{deg/pixel}. \quad (5)$$

## Mapping to momentum transfer

Mapping from scattering angles $(2\theta, \chi)$ to momentum transfer $\hbar\mathbf{q} = \hbar(\mathbf{k}_p - \mathbf{k}_s)$ proceeds in a system of spherical coordinates. The pump beam defines the polar axis along $\mathbf{k}_p = \omega_p/c\,\mathbf{e}_z$, where $c$ is the speed of light in vacuo and $\mathbf{e}_z$ the unit vector along $z$. The signal photon's $\mathbf{k}_s$ is fully determined by its magnitude $|\mathbf{k}_s| = \omega_s/c$ as well as the scattering angles $2\theta = \theta_s$ for its azimuth and

$$\chi \to \phi s = (\chi - \pi/2)(\sin 2\theta) - 1 + \pi/2 \quad (6)$$

for its polar angle. Following Eq. (3) [main text], $\mathbf{q}$ relates to the photonic level of the TLS via

$$\mathbf{k}^{\text{eff}}_{\gamma} = \mathbf{q} + \mathbf{G} = \mathbf{k}_p - \mathbf{k}_s + \mathbf{G} \quad (7)$$

analogous to the phase-matching condition. Writing the reciprocal lattice vector in the same coordinate system

$$\mathbf{G} = \omega_p c^{-1} \sin\theta_B \begin{pmatrix} 0 \\ \cos\theta B + \Delta\Omega \\ \sin\theta B + \Delta\Omega \end{pmatrix}, \quad (8)$$

we can compute $\mathbf{k}^{\text{eff}}_{\gamma}$ for all scattering angles and evaluate the pertaining dispersion relations $\omega_{\gamma}(\mathbf{q})$ or $E_{\pm}(\omega_{\gamma})$ numerically. The prefactor in Eq. (8) reflects Bragg's law.

## Background subtraction

The nonlinear conversion signal is imprinted on top of a stronger background, approximately within a ratio of 1/10. This background consists predominantly of (diffuse) Compton-scattering, which is further modulated by aberrations due to the crystal optics. To expose the XPDC signal, we subtract a scaled back-ground image from each recorded frame individually:

$$Frame_{corrected}^{(i)} = Frame^{(i)} - \left(Frame^{(offPM)} * \frac{\sum Frame^{(i)}}{\sum Frame^{(offPM)}}\right) \quad (9)$$

For that purpose we acquire a scattering image off the phase-matching condition, viz., $Frame^{(off\,PM)}$, that contains no XPDC, but still features the background with imprints of the analyzer. Scaling is based on the total (background) counts per frame, i.e., $Frame^{(i)}$, and varies slowly across the phase-matching range.

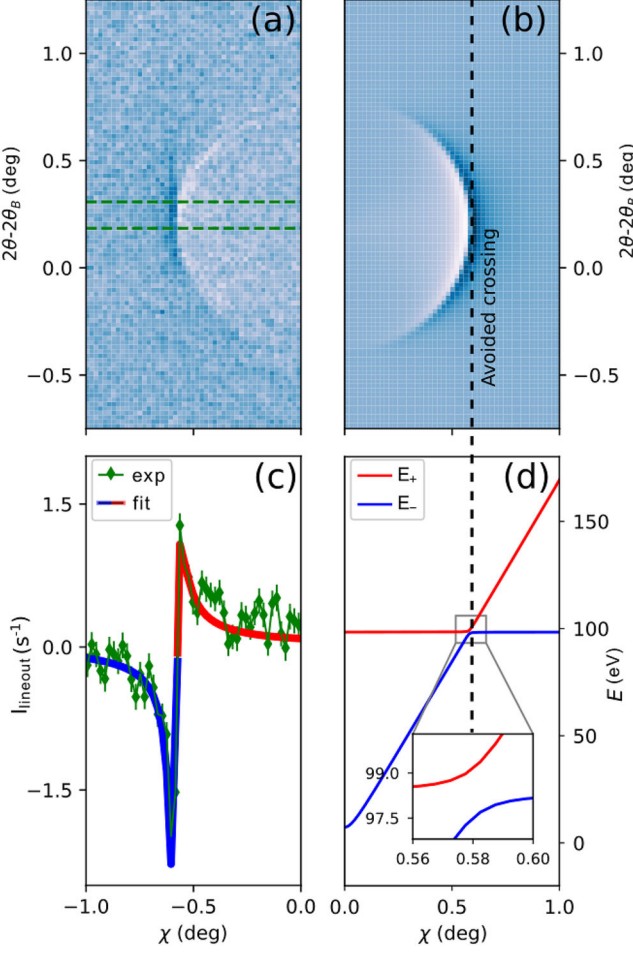

**Fig. 4 | Polaritonic features revealed by XPDC.** Comparison of **a** measured and **b** simulated scattering pattern for $\hbar(\omega_p - \omega_s) = 99.9$ eV on (111)-oriented diamond at $\Delta\Omega = 1.07°$. **c** Lineout of the experimental XPDC cone (green diamonds, with Poisson error) is taken around $2\theta\text{-}2\theta_B = 0.25°$ (region bounded by green dashed lines in a) to determine the coupling strength V by fitting the TLS-model (blue-red line). Color coding indicates the dominant scattering contribution from lower (blue) or upper (red) polariton branch. **d** Cross-section of the simulated polaritonic dispersion surfaces (also at $2\theta\text{-}2\theta_B = 0.25°$) visualize the anti-crossing's alignment with the scattering pattern (dashed vertical black line as guide to the eye). Inset highlights the level splitting of 2 $V = 1.64$ eV.

## Polarization-imprint on coupling V

We have postulated previously that parametric conversion from X-ray to optical photons will exhibit circumferential modulations of its scattering cones that relate to the polarization of the idler photons[24]. Confirming the same pattern experimentally in the EUV-case, we seek to incorporate this characteristic into our TLS-model as well. The transversality constraint from ref. 23 translates into our simplified coupling scheme as:

$$V \to V \times \left(1 - \left|\hat{\mathbf{k}}^{\text{eff}}_{\gamma} \cdot \hat{\mathbf{G}}\right|^2\right). \quad (10)$$

Angular variations derive from the orientation of the idler photon's wavevector $\mathbf{k}^{\text{eff}}_{\gamma}$ relative to the reciprocal lattice vector $\mathbf{G}$, represented through their respective unit vectors.

Using the modulation (10) in our simulations, we find excellent agreement with measured data in Fig. 4 [main text] (cf. minima around $\chi = 0$), thus establishing first evidence for a non-trivial polarization dependence also in the case of polaritonic XPDC. In the extended data below (Fig. 6), we illustrate how the polarization

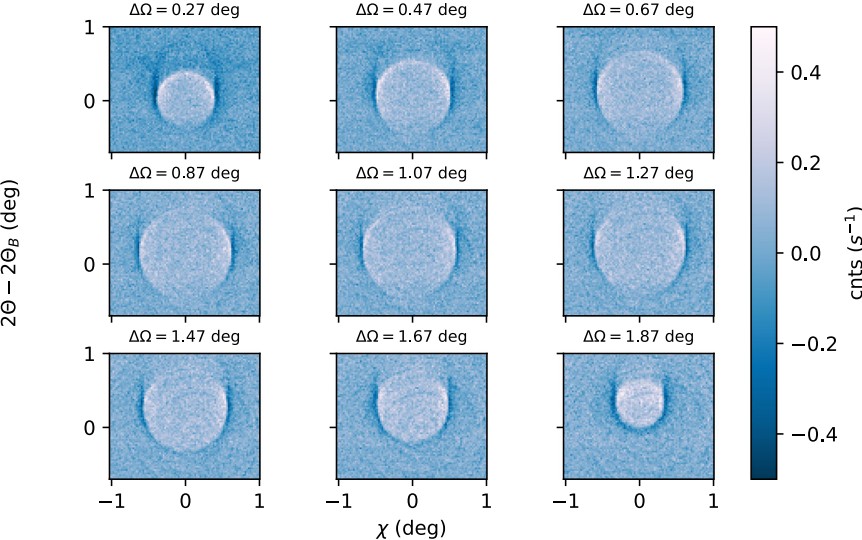

**Fig. 5 | Measurements of the XPDC cone extended.** All scattering images from Fig. 2d [main text] arranged in 2D[51]. Scattering angles ($2\theta, \chi$) are fixed to the same range for all frames; individual sample angle $\Delta\Omega$ is indicated above each plot.

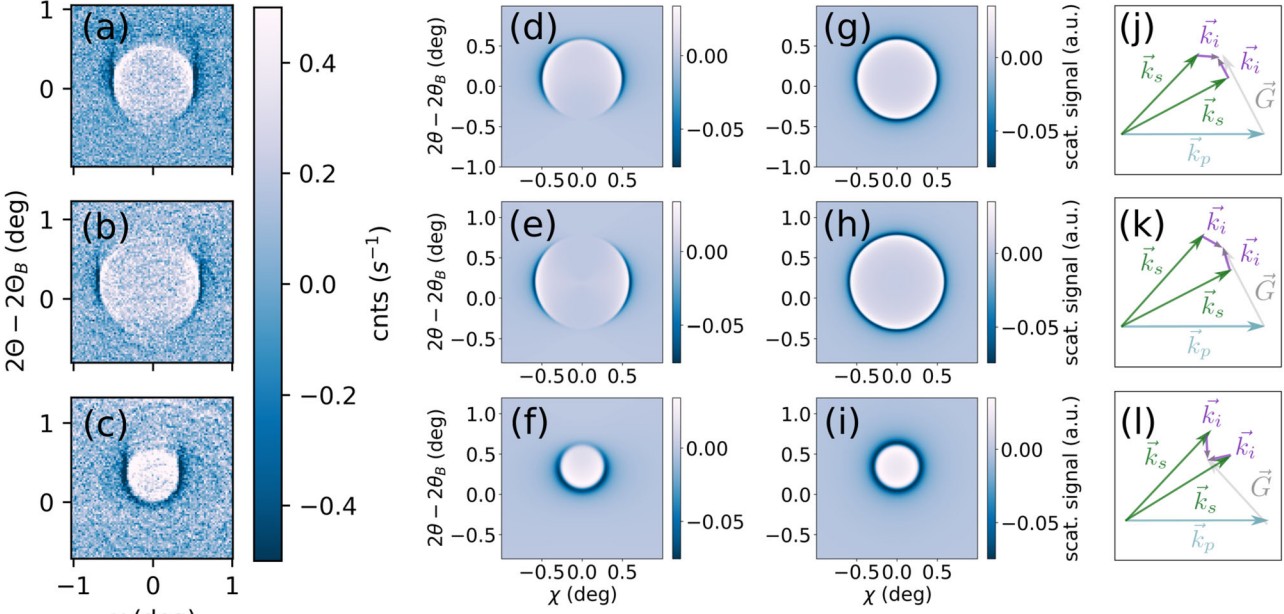

**Fig. 6 | Polarization imprint on the XPDC cone.** Juxtaposition of measured XPDC signal cones at different rocking angles (left column; **a–c**) with respective simulations that include (center-left; **d–f**) or disregard (center-right; **g–i**) polarization effects of the EUV-idler. The associated phase-matching conditions are sketched in the right-most column (**j–l**).

imprint can also be simulated for the other cases of Fig. 2a–c [main text]. While the first column reproduces the earlier experimental images (for sample detunings of $\Delta\Omega_a = 0.47°$, $\Delta\Omega_b = 1.07°$ and $\Delta\Omega_c = 1.87°$, respectively), the second column juxtaposes the TLS simulations with coupling according to Eq. (10). For comparison, we also include a simplified version of constant, isotropic coupling in the third column of extended data Fig. 6. Finally, its fourth column visualizes the phase-matching condition for each case. This influences the signal via Eq. (10), rendering it predominantly sensitive to EUV-quanta that are polarized in the same direction as the reciprocal lattice vector **G**. As such, all side-lobes of the XPDC cones (i.e., for phasematching $\mathbf{k}_\gamma^{\mathrm{eff}}$ out of plane) show pronounced signal strength, because there is a component of its polarization along **G** to be imaged. In contrast, the polariton's propagation direction ($\mathbf{k}_\gamma^{\mathrm{eff}}$) at

the upper or lower point of the cone may align with **G**, rendering its polarization largely orthogonal. This results in signal suppression, which is most visible in Fig. 2b around $\chi = 0$. Similarly, for the conditions shown in Fig. 2a or c, $\mathbf{k}_\gamma^{\mathrm{eff}}$ aligns with **G** at the lower- or uppermost phase-matching point, respectively. Again, this leads to signal suppression - while on each opposite side of the cone, the EUV-idler's polarization is allowed to be parallel to **G**, giving rise to strong signal and an overall horseshoe-shape.

The effect merits future investigation, concerning its interplay with more complex dielectric structures.

### Theoretical motivation of the TLS-model
To trace the imprint of polaritonic effects on XPDC, we base our description on the dynamic structure factor for inelastic X-ray scattering

(IXS)[37], which we express in atomic units below:

$$S(\mathbf{q},\omega) = \frac{1}{2\pi}\int_{-\infty}^{\infty}dt\,e^{i\omega t}\int d^3x\int d^3x'\,e^{-i\mathbf{q}\cdot(\mathbf{x}-\mathbf{x}')}\langle\hat{n}(\mathbf{x},t)\hat{n}(\mathbf{x}',0)\rangle. \quad (11)$$

Focusing on the density-density correlator and expanding its intermediate states, we write

$$\langle\hat{n}(\mathbf{x},t)\hat{n}(\mathbf{x}',0)\rangle \approx \sum_{n_i}^{VB}\sum_{\mathbf{k}_i}^{1.BZ} <\varphi_{n_i,\mathbf{k}_i}|\hat{n}(\mathbf{x})\sum_{n_m}^{CB}|\varphi_{n_m,\mathbf{k}_i}><\varphi_{n_m,\mathbf{k}_i}|<0|\hat{U}(t,0)$$
$$\sum_{n'_m}^{CB}|\varphi_{n_{m'},\mathbf{k}_i}>|0><\varphi_{n_{m'},\mathbf{k}_i}|\hat{n}(\mathbf{x}')|\varphi_{n_i,\mathbf{k}_i}>, \quad (12)$$

where we have expressed electronic states as Bloch-waves and–unconventionally for $S(\mathbf{q},\omega)$–included additional photonic degrees of freedom in terms of initially unoccupied Fock states $|0>$. Assuming phase-matching conditions and the dipole-approximation for the EUV-light-matter interaction[48] all terms remain approximately diagonal in their respective $\mathbf{k}$-space coordinates, which is reflected in the single summation over $\mathbf{k}_i$. The central matrix element $<\varphi_{n_m,\mathbf{k}_i}|<0|\hat{U}(t,0)|\varphi_{n_{m'},\mathbf{k}_i}>|0>$ consists of highly excited electronic states, which would conventionally be considered only for their role as Compton-electrons–evolving quasi-free at -100 eV. Within the band structure, however, such an electron also has a finite chance of returning to its initial valence-band state (now a hole at $\varphi_{n_i,\mathbf{k}_i}$) by emission of a 100 eV photon. This photonic coupling of states gives rise to the EUV-polariton. Notwithstanding the large number of overall bands $n_m$, there will typically be only very few (if any) states that match the photonic resonance at 100 eV $= \epsilon_{n_m,\mathbf{k}_i} - \epsilon_{n_i,\mathbf{k}_i}$. For our modeling, we will focus on those pairs of states $\varphi_{n_m,\mathbf{k}_i}; \varphi_{n_i,\mathbf{k}_i}$, that are closest to the transition at each $\mathbf{k}$-point. This effectively establishes a set of near-resonant two-level systems (TLS) with Hamiltonians deriving from

$$\hat{H} = \hat{H}_{Bloch} + \hat{H}_{phot} + \hat{H}_{interaction} \quad (13)$$

$$\Rightarrow \begin{pmatrix} \epsilon_{n_i,\mathbf{k}_i} & 0 \\ 0 & \epsilon_{n_m,\mathbf{k}_i} \end{pmatrix} + \begin{pmatrix} \omega_{\mathbf{k}_\gamma} & 0 \\ 0 & 0 \end{pmatrix} + \begin{pmatrix} 0 & V(n_i,\mathbf{k}_i) \\ V^\dagger(n_i,\mathbf{k}_i) & 0 \end{pmatrix}, \quad (14)$$

including coupling elements $V(n_i,\mathbf{k}_i) = \sum_{\mathbf{k}_\gamma}^{PM}\sum_\lambda\sqrt{\frac{2\pi}{V_\diamond\omega_{\mathbf{k}_\gamma}}}<\varphi_{n_i,\mathbf{k}_i}|\epsilon^*_{\mathbf{k}_\gamma,\lambda}\hat{\mathbf{p}}|\varphi_{n_m,\mathbf{k}_i}><1|\hat{a}^\dagger_{\mathbf{k}_\gamma,\lambda}|0>$ based on the interaction terms of ref. [48], with the range of photon momenta being constraint by phase-matching (PM). Equivalent reductions to resonant few-level emitters are at the core of many polaritonic descriptions–typically leading to Tavis-Cummings or Dicke-type models (see, e.g., ref. [14]). We choose to diagonalize each $\mathbf{k}$-point and frequency individually, leading to the well-known eigenenergies

$$E_\pm(n_i,\mathbf{k}_i) = \frac{(\epsilon_{n_m,\mathbf{k}_i} - \epsilon_{n_i,\mathbf{k}_i}) + \omega_{\mathbf{k}_\gamma}}{2}$$
$$\pm\sqrt{\frac{1}{4}((\epsilon_{n_m,\mathbf{k}_i} - \epsilon_{n_i,\mathbf{k}_i}) - \omega_{\mathbf{k}_\gamma})^2 + |V(n_i,\mathbf{k}_i)|^2}. \quad (15)$$

The corresponding eigenstates are obtained by the transformation

$$\hat{T}(n_i,\mathbf{k}_i)|\varphi_{n_m,\mathbf{k}_i}>|0> = \sum_\pm c_{e\pm}(n_i,\mathbf{k}_i)|\phi_\pm^{pol}(n_i,\mathbf{k}_i)>$$
$$\text{with } c_{e\pm}(n_i,\mathbf{k}_i) = \left(1 + \frac{|V(n_i,\mathbf{k}_i)|^2}{(E_\pm(n_i,\mathbf{k}_i)-\omega_{\mathbf{k}_\gamma})^2}\right)^{-\frac{1}{2}}, \quad (16)$$

which also transforms the original matrix element into

$$<\varphi_{n_m,\mathbf{k}_i}|<0|\hat{U}(t,0)|\varphi_{n_m,\mathbf{k}_i}>|0> = \sum_\pm|c_{e\pm}(n_i,\mathbf{k}_i)|^2 e^{-iE_\pm(n_i,\mathbf{k}_i)t} \quad (17)$$

Combining Eqs. (1), (2) and (7), the dynamic structure factor near phase-matching (PM) becomes

$$S^{pol}(\mathbf{q},\omega) = \frac{1}{V_\diamond}\sum_{n_i}^{VB}\sum_{\mathbf{k}_i}^{1.BZ}\left(\left|\int d^3x\,e^{-i\mathbf{q}\mathbf{x}}<\varphi_{n_i,\mathbf{k}_i}|\hat{n}(\mathbf{x})|\varphi_{n_m,\mathbf{k}_i}>\right|^2\right.$$
$$\left.\sum_{\mathbf{k}_\gamma}\sum_\pm|c_{e\pm}(n_i,\mathbf{k}_i)|^2\delta(E_\pm(n_i,\mathbf{k}_i)-\omega)\right) \quad (18)$$

This represents the sum of partial scattering factors from all initial electronic states $\varphi_{n_i,\mathbf{k}_i}$ – each dressed by a TLS, subject to energy conservation. Interpreting the summation as an average, we approximate the result as a single, effective TLS that dresses the overall dynamic structure factor with its Hopfield coefficients $c_{e\pm}$:

$$S^{pol}(\mathbf{q},\omega) \approx S^{IXS}(\mathbf{q},\omega)\cdot\sum_\pm|c_{e\pm}|^2 e^{-(E_\pm-\omega)^2/2\Gamma_{in}^2} \quad (19)$$

The single, effective two-level system marks the starting point for our main discussion. It is governed by the Hamiltonian of Eq. (M8), where $\omega_e$ is the average transition energy and $\omega_\gamma$ the average frequency of coupled photonic modes. Any relative variation within a sector is taken to be small and accounted for by an intrinsic broadening factor $\Gamma_{in}$, while the overall coupling of electronic and photonic sectors is now mediated by the collective coupling strength $V$.

## Scattering yield from TLS-model

In order to connect our effective model to the observable scattering yield, we first relate $S^{pol}(\mathbf{q},\omega)$ to the double-differential scattering cross section of regular inelastic X-ray scattering (IXS)[40] as follows

$$\frac{d\sigma^{pol}}{d\Omega_s d\omega_s} = \left(\frac{d\sigma}{d\Omega_s}\right)_{Th}\frac{\omega_s}{\omega_p}S^{pol}(\mathbf{q},\omega_p-\omega_s), \quad (20)$$

with the Thomson cross section $(...)_{Th}$ governing the proportionality[40]. After convoluting Eq. (20) with the incident fluence as well as the (Gaussian-modeled) transmission function of the analyzer-setup, we integrate for the scattering yield per pixel (i.e., covering (20.5 mdeg)$^2$):

$$Y_{pix} = p\cdot\sum_\pm|c_{e\pm}|^2 e^{-(\hbar\Delta\omega_{an}-E_\pm)^2/2(\hbar\Gamma)^2}. \quad (21)$$

Both $E_\pm$ and $c_{e\pm}$ are implicit functions of the scattering angles $2\theta$ and $\chi$ at each point of evaluation. The overall bandwidth $\Gamma$ as well as the energy-loss determined by the analyzer $\Delta\omega_{an} \sim \omega_\gamma$ are left as fitting parameters for simplicity, while the overall prefactor $p$ is obtained from intrinsic background scattering (see below). Note that we have re-instated $\hbar$ starting from Eq. (21).

## Fitting procedure

We, consistently, resort to background-subtracted data for fitting, using $Y_{fit} = Y_{pix} - Y_{bg}$. The pertinent background consists of inelastic (Compton) scattering and is captured by our Eq. (8) in the limit $S^{pol}(\mathbf{q},\omega) \to S^{IXS}(\mathbf{q},\omega)$. For regions of $\mathbf{q}$-space *away* from polaritonic phase-matching. This translates onto Eq. (21) as

$$\lim_{n\to\infty}Y_{pix}(\omega_\gamma) = Y_{BG} = pe^{-(\Delta\omega_{an}-\omega_e)^2/2\Gamma^2}. \quad (22)$$

and automatically determines the prefactor $p$.

Binning the 2D data in a line-out along $\chi$ of width 7 pixel [see main text], we fit $Y_{fit}$ using the curve fit-routine from the freely-available python-package scipy.optimize. The result is shown in Fig. 4 [main text], with the underlying parameters determined to be: $\hbar\omega_e = 98.36 \pm 0.39$ eV, $V = 0.82 \pm 0.08$ eV, $\hbar\Gamma = 1.64 \pm 0.22$ eV and for the setup $\hbar\Delta\omega_{an} = 99.90 \pm 0.22$ eV. The latter refines our knowledge of the difference $\hbar(\omega_p - \omega_s) \approx 100$ eV. We assess the quality of the fitting by estimating the uncertainty by its reduced $X^2 \approx 2.1$, which is based on the counting error and indicates good agreement.

We note that the line shape could also be fitted by the Fano-formula[49] on a phenomenological basis. This was done by Tamasaku et al. under similar conditions, suggesting an interference between Compton scattering and XPDC[35]. However, these processes should not interfere quantum mechanically (see also Tamasaku et al.[50]), which renders a Fano-model inapplicable. In contrast, our polaritonic interpretation reconciles both processes and yields Eq. (15) to describe the full, hybridized phenomenon.

## Data availability

The experimental data generated in this study have been deposited in the Zenodo repository under accession code https://doi.org/10.5281/zenodo.15282658.

## Code availability

Codes used to generate the simulation results in this study are available upon reasonable request to the corresponding authors.

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

## Acknowledgements
The authors acknowledge the European Synchrotron Radiation Facility (ESRF) for provision of synchrotron radiation under proposal number HC-4489 and HC-4907 using beamline ID20. This work is supported and partly funded by the Cluster of Excellence "Advanced Imaging of Matter" of the Deutsche Forschungsgemeinschaft (DFG)—EXC 2056—project ID 390715994 (D.K., C.B., F.K., N.R.). Parts of this work are funded through the Max Planck School of Photonics supported by BMBF, Max Planck Society, and Fraunhofer Society (D.K., N.R.). Parts of this work are funded through the German-Israeli Foundation for Scientific Research and Development under GIF Grant No: I-1495-303.7/2019 Project (N.R.). In addition, this study was supported by the Research Council of Finland grants 295696 and 303914 (S.H.). D.K. and C.B. thank Ralf Röhlsberger for discussions and support during the completion of this manuscript.

## Author contributions
D.K. and C.B conceived the experiment with N.R. hosting the project. D.K., C.B., C.J.S., B.D. performed the experiments together with F.K. and S.H. C.B. analyzed the experimental data with support from X.B. D.K. developed the theoretical description and F.K. performed its modeling and fitting. D.K., C.B., N.R., C.J.S. and S.H. interpreted the data, while all authors discussed the results. D.K. and C.B. wrote the manuscript with contributions from all authors.

## Funding

## Competing interests
The authors declare no competing interest.
