## [Transparent Peer Review file · Nature Communications]

X-ray parametric down-conversion reveals EUV-polariton

Corresponding Author: Dr Christina Boemer

Version 0:

Reviewer comments:

Reviewer #1

(Remarks to the Author)

The authors report the first experimental record of the full down-conversion cone at X-ray wavenegths. This is based on a smart modification of the analyser used in a typical down-conversion experiment (performed at the European Synchrotron Radiation Facility), that allows to collect signal photons at all angles, while idler photons remain unobserved. These new capabilities reveal a somehow unexpected picture, of a double cone instead of a single one, which the authors attribute to the excitation of an X-ray polariton.

The experimental work is interesting and innovative, and could justify publication in Nature Communications. Practical set-ups for nonlinear processes such as parameteric down-conversion are relatively scarce in the X-ray regime, but can have significant impact on applications in quantum technologies. The experiment, based on X-ray diffraction by a common crystal and a smart yet simple modification of the experimental set-up and the data processing sequence has the potential to facilitate more intense exploration of this regime, and, in my opinion, this is where the true novelty and impact of the work lies.

On the other hand, at this stage, I do not find the discussion and proposed explanation of the data based on the excitation of a UV polariton convincing, and this is where the revision should focus, possibly also with further experiments.

In particular:

1) The authors rely on the possibility of a vague virtual electronic excitation, which, additionally, can be triggered several times by the same signal photon. This might indeed be true, but in the present version of the manuscript this remains simply speculation, and even the energy of this state is obtained by fitting. More arguments/evidence that such a virtual state can be excited are necessary.

2) The emergence of the polariton is analysed in a phenomenological way, based on the simple two-level Rabi system (or, equivalently, the coupled-harmonic-oscillators model), without any damping rates included. My first objection to this approach concerns exactly the damping, and the conclusion of having achieved strong coupling based on the width of the anticrossing and the linewidth of the *observed scattering yield*. But the true definition of strong coupling relies on the two systems (here photons and electrons) exchanging energy faster than they lose it to their own decay channels, meaning that the comparison should somehow be made between the width of the anticrossing, and the linewidths of the two individual coupled states---see for example the long discussion on coupling criteria in Torma & Barnes - Rep. Prog. Phys. 78, 013901 (2015).

A second objection concerns the description based on a single two-level system. Based on the sketch of Fig. 3a, one would anticipate that it is several virtual transitions that participate simultaneously in the coupling. This, then, should be either described with a Dicke-like model [Dicke - Phys. Rev. 93, 99 (1954)] in the case of fermions, or with a bosonic description in the case where all these states form a collective dipole moment together [see e.g. Tserkezis et al. - Rep. Prog. Phys. 83, 082401 (2020)].

3) Similarly, if the polaritonic description is to be adopted, the authors should provide some guidelines for calculating (at a first stage in a simplified picture) the coupling strength V --which again relates to specifying the actual nature of the transition that takes place.

4) In any case, the narrowness of the splitting, and the shape of the data in Fig. 4c, suggest something closer to a Fano resonance as a result of this narrow transition (let's assume it is indeed a virtual electronic transition) over a wider photonic

background. I wonder if the data of Fig. 4c would not be better reproduced by the Fano formula [Fano - Phys. Rev. 124, 1866 (1961)].

As a couple of suggestions for further work that could shed light on these issues, I wonder if it would be possible to perform the same experiment with a different material, other than diamond, or with a different crystallographic facet of the same sample, so as to pick a different reciprocal-lattice vector, and thus generate different signal/idler photons. In a similar direction, what would happen with larger rotations of the same sample?

Overall, I believe that the experiment itself could justify publication in Nature Communications, but if the authors want to insist on the picture of the UV polariton, more evidence and work is needed.

Reviewer #2

(Remarks to the Author)

The article "X-ray parametric down-conversion reveals EUV-polariton" is a noteworthy work in the field of X-ray parametric down conversion. The significance/advancement is mainly regarding the identification of the EUV-polariton mechanism. This is interesting to researchers in the field, as the experimental evidence presented herein helps solidify the mechanism of the X-ray PDC and, as the authors note, opens up new research avenues. Combining the experimental results with the TLS model bolsters the claim and helps to support the conclusions of the paper. I do not find any flaws in the analysis, interpretation, or conclusions, and the methodology seems sound. My overall recommendation is acceptance with some relatively minor revisions.

Regarding claims of resolving the XPDC cone for the first time, e.g. on:

-> Lines (25)-(26): "Here, we resolve the full down-conversion cone at x-ray 25 wavelengths for the first time [...]"

-> Line (47): "This enables us to resolve the conversion cone of XPDC for the first time [...]"

-> Line (59)-(60): "Remarkably, a full signal cone has never before been observed in the x-ray regime [...]"

-> Line (90)-(91): "The introduction of 2D imaging fundamentally distinguishes our setup from all previous approaches to measure XPDC [...]" (and other places)

I do not believe this work constitutes the first spatially resolved/2D imaging of cone measurements of the XPDC process. I would draw the author's attention to arXiv:2310.13078, recent work from the NSLS-II, which also claims to resolve the down conversion cone with an area detector for the first time. I would encourage the authors to consider/cite their work and either (1) remove claims of the 'first time' etc. or, if they feel the referenced work does not fully meet the claim but their work does (2) address this point explicitly.

On a related note, the authors claim (emphasis mine) to "resolve the **full** down-conversion" in the abstract, but then later on lines (67-68) specify: "We focus specifically on down-conversion into signal photons [...] while the idler remains unobserved". I find the language of saying the "full" down-conversion is resolved' at odds with the fact that the lower energy photon is not observed in these measurements. I would recommend rewriting/clarification.

These issues notwithstanding, the work presented is of high quality, interesting to researchers in the field, and is recommended for publication.

Reviewer #3

(Remarks to the Author)

The development and implementation of non-linear techniques at higher photon energies is an important area of intense research. One approach is XPDC. In the present manuscript, the authors report a remarkable progress in this direction. They observe the full XPDC cone at once for the first time, explore its shape across the entire XPDC ellipsoid, and identify the formation of an EUV polariton. Strikingly, the analysis reveals that it is naturally close to the strong-coupling limit. The comprehensive experimental data is convincingly supported by adequate theory modelling. These developments are of high and broad significance, and open a clear path to a broad range of interesting follow-up works, as hinted at in the discussion. In my view, these results fully merit a publication in Nature Communications.

Before publication, I have few questions:

* Is the "universality of polaritons" stressed in the introduction really an important aspect of the work? Theoretically, it is not very surprising?

* What is the overall conversion efficiency of incident photon into XPCS photon (created / detected)? What is the loss due to the energy analysis / imaging? How does the overall efficiency compare to the previous experiments, given that the authors claim superior data-acquisition efficiency?

* Is Bragg scattering an issue? How does it compare to the Compton scattering background? Or is it fully removed by the analyzer?

* Throughout the analysis, the authors claim "Confirming this pattern will allow us to unambiguously identify XPDC without coincident detection of any idler photon". It seems perfectly convincing, and I do not argue against it - in particular, given the good agreement with the later theory analysis. But is there actually a theoretical support or even a proof behind this statement?

* Which part ("radius"/ opening angle) of the polariton structure is chosen to compare with the theoretical XPCS cone opening angles?

* Fig 2(c) shows a horseshoe-structure of reduced counts, rather than the double-lobes in (b/c). Can this be understood from the polarization argument?

* What determines the value of V / Γ ? Is there a way to tune their ratio (e.g., via the sample temperature, G vector, idler frequency, ...?)

* What are the prospects of generalizing the scheme beyond EUV energies?

* as a technical remark: the subindices "i" on the idler wave vector are different in text and figures, and different to that on ω_i

Version 1:

Reviewer comments:

Reviewer #1

(Remarks to the Author)

The authors have revised the manuscript to address all referee comments. I rather appreciate the length of the responses in the rebuttal, and the incorporation of the response to my first comment in the Methods section. I believe that the manuscript can now be accepted.

Reviewer #2

(Remarks to the Author)

I believe that the comments from the authors adequately respond to the concerns raised by myself and the other reviewers. I recommend the article for publication.

Reviewer #3

(Remarks to the Author)

In their revised manuscript version, the authors have resolved all issues raised by the referees. My positive assessment of the significance of the results and their expected impact remains unchanged.

The additional theoretical background material provided to the questions is convincing. I do not think that it is a major drawback at this stage that not all aspects of the experiment can be understood quantitatively from the theory. It is more important that the provided models support the discussion and interpretation in the manuscript sufficiently, and in my view this is the case.

I appreciate the careful and detailed answer on the efficiency - indeed it is very important to clarify exactly what is meant by that in order to make the work comparable (which is sometimes difficult in the x-ray xpdc literature for the readers).

Overall, the author's response also highlights the potential for future research enabled by the present study, which is likely to attract interest of a broad readership. One crucial aspect is the value of V or its ratio to Γ and the authors argue convincingly that their setup may allow for testing this idea in future experiments.

I therefore recommend publication of the present manuscript as is.

RESPONSE TO REVIEWER COMMENTS

on the manuscript 'X-ray parametric down-conversion reveals EUV-polariton'

First and foremost, we would like to thank all reviewers for their kind consideration and constructive feedback in reviewing our manuscript. We are grateful for their thoroughly positive evaluations and appreciate their many helpful questions and comments. In the following, we will address all remarks on a point-by-point basis - quoting each individual comment in **blue** and answering in **black** font-color. Where necessary, we apply changes to our manuscript accordingly and highlight them in a separate document. For convenience, we have added all required references at the end of this document, while keeping their enumeration congruent with the updated manuscript. Two additional references, which are exclusive to this comment are denoted C1 and C2.

Reviewer #1 (Remarks to the Author):

The authors report the first experimental record of the full down-conversion cone at X-ray wavelengths. This is based on a smart modification of the analyser used in a typical down-conversion experiment (performed at the European Synchrotron Radiation Facility), that allows to collect signal photons at all angles, while idler photons remain unobserved. These new capabilities reveal a somehow unexpected picture, of a double cone instead of a single one, which the authors attribute to the excitation of an X-ray polariton.

The experimental work is interesting and innovative, and could justify publication in Nature Communications. Practical set-ups for nonlinear processes such as parametric down-conversion are relatively scarce in the X-ray regime, but can have significant impact on applications in quantum technologies. The experiment, based on X-ray diffraction by a common crystal and a smart yet simple modification of the experimental set-up and the data processing sequence has the potential to facilitate more intense exploration of this regime, and, in my opinion, this is where the true novelty and impact of the work lies.

We thank the reviewer for their positive assessment of our results and their appreciation of the experimental opportunities opened by the setup. We could not agree more strongly on the wish to intensify the exploration of parametric down-conversion and quantum optics in the x-ray regime. With current and future studies, we are actively pursuing this direction ourselves as well.

On the other hand, at this stage, I do not find the discussion and proposed explanation of the data based on the excitation of a UV polariton convincing, and this is where the revision should focus, possibly also with further experiments.

In particular:

1) The authors rely on the possibility of a vague virtual electronic excitation, which, additionally, can be triggered several times by the same signal photon. This might indeed be true, but in the present version of the manuscript this remains simply speculation, and even the energy of this state is obtained by fitting. More arguments/evidence that such a virtual state can be excited are necessary.

We thank the reviewer for their critical comment and understand the need for clarification on the electronic state(s), which contribute to the EUV-polariton. In our original manuscript, we adopted the notion of a 'virtual electronic excitation' to summarize the effect of several possible excitations. This was intended to simplify the overall discussion – in line with our effective two-level model. At the same time, we understand that this simplification on its own could also be perceived as vague. In particular, we note that the wording 'virtual' was an unfortunate misnomer on our side and should rather have read 'effective' throughout the manuscript. As a matter of fact, the electronic states involved in the observed polaritonic excitation are not intrinsically virtual, but instead comprise a range

of *real* states within the band structure of diamond, far above its ionization threshold (cf. Fig. 1 below). These states become intermediately excited, if they can be addressed by the energy transfer of $\sim 100 \text{ eV} = E_p - E_s$ that is experimentally fixed by the analyzer's transmission relative to the incident beam.

Notably – within the admissible bandwidth of our setup – these physically real states will exhibit various energies, which we do not determine individually. Instead, we opt to summarize their effect into the aforementioned ‘effective’ excitation, for which we fit the apparent average energy. We did not intend for this procedure to be vague, but rather to yield a simplification, which allows us to focus on the phenomenological discussion of our observation.

Moreover, this simplification does not affect the fundamental existence of an EUV-polariton, as either a single *effective* or several *real* electronic states will hybridize with the electromagnetic field into some kind of polaritonic excitation. This behavior is universal to a degree that reviewer #3 even considered our statement of this as ‘not very surprising’.

While we, thus, stand by the results of our phenomenological two-level model and cherish its simplicity for the interpretation of our observations, we also want to take this opportunity raised by the reviewer to augment our discussion by giving a broader theoretical motivation of that model. An outline of this will be included with the revised methods section (6) of the manuscript as well.

Theoretical motivation of the TLS-model:

In order to trace the imprint of polaritonic effects on XPDC, we can start out from the dynamic structure factor for general inelastic x-ray scattering [37]:

$$S(\mathbf{q}, \omega) = \frac{1}{2\pi} \int_{-\infty}^{\infty} dt e^{i\omega t} \int d^3x \int d^3x' e^{-i\mathbf{q}\cdot(\mathbf{x}-\mathbf{x}')} \langle \hat{n}(\mathbf{x}, t) \hat{n}(\mathbf{x}', 0) \rangle. \quad (1)$$

In here, we focus on the density-density correlator

$$\langle \hat{n}(\mathbf{x}, t) \hat{n}(\mathbf{x}', 0) \rangle = \langle I | \hat{n}(\mathbf{x}) \hat{U}(t, 0) \hat{n}(\mathbf{x}') | I \rangle \quad (2)$$

$$= \langle I | \hat{n}(\mathbf{x}) \sum_M |M\rangle \langle M| \hat{U}(t, 0) \sum_{M'} |M'\rangle \langle M'| \hat{n}(\mathbf{x}') | I \rangle, \quad (3)$$

which can be written as an expectation value in the initial state $|I\rangle$ and expanded using intermediate states $|M\rangle, |M'\rangle$. Usually, all of these states are assumed to be purely electronic, to compute the dynamic structure factor. Likewise, the Hamiltonian \hat{H} generating the time evolution $\hat{U}(t, 0)$ is usually taken as $\hat{H} = \hat{H}_{elec}$. Only. Instead in our case, we will consider both electronic *and* photonic states - hybridizing into the EUV-polariton under the action of the combined Hamiltonian

$$\hat{H} = \hat{H}_{elec} + \hat{H}_{phot} + \hat{H}_{interaction}. \quad (4)$$

For simplicity, we restrict our present discussion of the electronic system to a Bloch-type band structure, i.e.,

$$\hat{H}_{elec} \approx \hat{H}_{Bloch} = \sum_n^{\text{bands}} \sum_{\mathbf{k}}^{1.\text{BZ}} \epsilon_{n,\mathbf{k}} \hat{c}_{n,\mathbf{k}}^\dagger \hat{c}_{n,\mathbf{k}}$$

with the single particle states corresponding to the energies $\epsilon_{n,\mathbf{k}}$ being

$$\varphi_{n,\mathbf{k}}(\mathbf{x}) = \frac{1}{\sqrt{V_0}} e^{i\mathbf{k}\mathbf{x}} u_{n,\mathbf{k}}(\mathbf{x})$$

Above, n denotes individual bands and \mathbf{k} enumerates distinct points within the first Brillouin zone (1. BZ), at a mode volume of $\frac{(2\pi)^3}{V_0}$. For the electromagnetic field, we adopt the description of Ref. [50], taking

$$\hat{H}_{phot} = \sum_{\mathbf{k}, \lambda} \omega_{\mathbf{k}} \hat{a}_{\mathbf{k}, \lambda}^\dagger \hat{a}_{\mathbf{k}, \lambda}; \quad \omega_{\mathbf{k}} = c|\mathbf{k}|,$$

where \mathbf{k} and λ denote the wave vector and polarization state of each photonic mode, respectively. Individual Fock-states of the electromagnetic field $|\dots n_{\mathbf{k},\lambda} \dots\rangle$ can be occupied by $n_{\mathbf{k},\lambda}$ photons each and acted upon by creation (annihilation) operators $\hat{a}_{\mathbf{k},\lambda}^\dagger$ ($\hat{a}_{\mathbf{k},\lambda}$).

The coupling of electronic and photonic system reads

$$\hat{H}_{\text{interaction}} = \hat{\mathbf{p}} \cdot \sum_{\mathbf{k},\lambda} \sqrt{\frac{2\pi}{V\omega_{\mathbf{k}}}} (\hat{a}_{\mathbf{k},\lambda} \boldsymbol{\epsilon}_{\mathbf{k},\lambda} + \hat{a}_{\mathbf{k},\lambda}^\dagger \boldsymbol{\epsilon}_{\mathbf{k},\lambda}^*), \quad (6)$$

where we have used the dipole-approximation to simplify upon Ref. [50]. Further, V gives the quantization volume and $\boldsymbol{\epsilon}_{\mathbf{k},\lambda}$ marks the polarization vector per mode.

It is imperative to note that all Hamiltonians are diagonal in their respective \mathbf{k} -space coordinates. This will render our problem mostly separable, as we can see by re-writing Eq. (3) in terms of electronic and photonic states (e.g., $|M\rangle \rightarrow |\varphi_m, n_m^{\text{phot.}} = 0\rangle = |\varphi_{n_m, \mathbf{k}_m}\rangle \times |0\rangle$):

$$\begin{aligned} & \langle I | \hat{n}(\mathbf{x}) \sum_M |M\rangle \langle M| \hat{U}(t, 0) \sum_{M'} |M'\rangle \langle M' | \hat{n}(\mathbf{x}') |I\rangle \\ & \rightarrow \sum_i \langle \varphi_i, 0 | \hat{n}(\mathbf{x}) \sum_m |\varphi_m, 0\rangle \langle \varphi_m, 0 | \hat{U}(t, 0) \sum_{m'} |\varphi_{m'}, 0\rangle \langle \varphi_{m'}, 0 | \hat{n}(\mathbf{x}') | \varphi_i, 0\rangle \\ & = \sum_{n_i}^{\text{VB}} \sum_{\mathbf{k}_i}^{\text{1.BZ}} \langle \varphi_{n_i, \mathbf{k}_i} | \hat{n}(\mathbf{x}) \sum_{n_m}^{\text{CB}} \sum_{\mathbf{k}_m}^{\text{1.BZ}} |\varphi_{n_m, \mathbf{k}_m}\rangle \langle \varphi_{n_m, \mathbf{k}_m} | \langle 0 | \hat{U}(t, 0) \sum_{n_{m'}}^{\text{CB}} \sum_{\mathbf{k}_{m'}}^{\text{1.BZ}} |\varphi_{n_{m'}, \mathbf{k}_{m'}}\rangle |0\rangle \\ & \quad \times \langle \varphi_{n_{m'}, \mathbf{k}_{m'}} | \hat{n}(\mathbf{x}') | \varphi_{n_i, \mathbf{k}_i}\rangle. \end{aligned} \quad (7)$$

Analogous to the regular treatment of $S(\mathbf{q}, \omega)$, there are no EUV photons present initially (cf., photonic state $|0\rangle$), but can be created within $\hat{U}(t, 0)$. Recalling that this does not change the electronic momentum \mathbf{k} substantially, we can fix $\mathbf{k}_m = \mathbf{k}_{m'}$. Moreover, we know that polaritonic signatures only emerge at phase-matching conditions, i.e., close to Bragg peaks, where the momentum transfer onto the electronic system is commensurate with the reciprocal lattice $\mathbf{q} \approx \mathbf{G}$. Within the first Brillouin-zone (i.e., modulo any \mathbf{G}), this entails that also $\mathbf{k}_i \approx \mathbf{k}_m = \mathbf{k}_{m'}$ - which further simplifies Eq. (8) into

$$\begin{aligned} & \sum_{n_i}^{\text{VB}} \sum_{\mathbf{k}_i}^{\text{1.BZ}} \langle \varphi_{n_i, \mathbf{k}_i} | \hat{n}(\mathbf{x}) \sum_{n_m}^{\text{CB}} |\varphi_{n_m, \mathbf{k}_i}\rangle \langle \varphi_{n_m, \mathbf{k}_i} | \langle 0 | \hat{U}(t, 0) \sum_{n_{m'}}^{\text{CB}} |\varphi_{n_{m'}, \mathbf{k}_i}\rangle |0\rangle \langle \\ & \varphi_{n_{m'}, \mathbf{k}_i} | \hat{n}(\mathbf{x}') | \varphi_{n_i, \mathbf{k}_i}\rangle \end{aligned} \quad (9)$$

Now, it is time to analyze the central matrix element

$$\langle \varphi_{n_m, \mathbf{k}_i} | \langle 0 | \hat{U}(t, 0) | \varphi_{n_{m'}, \mathbf{k}_i}\rangle |0\rangle. \quad (10)$$

It consists of highly excited electronic states, which would usually evolve as quasi-free Compton-electrons in the diagonal Bloch-Hamiltonian at $\epsilon_{n_m, \mathbf{k}_i} \approx 100$ eV. Nonetheless, the electron is still part of the diamond band structure and thereby has a small chance of returning to its initial valence-band state (now a hole at $\varphi_{n_i, \mathbf{k}_i}$) by emission of a 100 eV photon (see Fig. 1). Note, that this is only possible for the fraction of electrons, which have undergone near-vertical excitation and did not change their momentum via subsequent scattering events either. Otherwise electrons will not contribute coherently to the polaritonic excitation, but only to the incoherent background.

Fig. 1: Photon absorption (and conversely re-emission) pathways within the band structure of diamond at 100 eV for different initial valence states (DFT simulation). The inset shows a zoom-in around 100 eV, illustrating the scarce occurrence of sharp resonance conditions (arrows), which motivates a two-level approximation in their vicinity.

While there are several bands n_m for potential excitations to be found in the region from 90 – 110 eV, we will focus our description on those pairs of states $\varphi_{n_m, \mathbf{k}_i}$; $\varphi_{n_i, \mathbf{k}_i}$, that are closest to the photonic resonance $100 \text{ eV} = \epsilon_{n_m, \mathbf{k}_i} - \epsilon_{n_i, \mathbf{k}_i}$ and will therefore contribute dominantly to the EUV-polariton. Equivalent reductions to resonant few-level emitters are at the core of many polaritonic descriptions - leading to Tavis-Cummings or Dicke-type models (see, e.g., Ref. [14]).

Effectively, this treatment establishes a near-resonant two-level system (TLS) for each \mathbf{k} -point with the Hamiltonian deriving from

$$\hat{H} = \hat{H}_{\text{Bloch}} + \hat{H}_{\text{phot}} + \hat{H}_{\text{interaction}} \quad (11)$$

$$\Rightarrow \begin{pmatrix} \epsilon_{n_i, \mathbf{k}_i} & 0 \\ 0 & \epsilon_{n_m, \mathbf{k}_i} \end{pmatrix} + \begin{pmatrix} \omega_{\mathbf{k}_\gamma} & 0 \\ 0 & 0 \end{pmatrix} + \begin{pmatrix} 0 & V(n_i, \mathbf{k}_i) \\ V^\dagger(n_i, \mathbf{k}_i) & 0 \end{pmatrix}. \quad (12)$$

Here, $V(n_i, \mathbf{k}_i) = \sum_{\mathbf{k}_\gamma}^{PM} \sum_{\lambda} \sqrt{\frac{2\pi}{V\omega_{\mathbf{k}_\gamma}}} \langle \varphi_{n_i, \mathbf{k}_i} | \epsilon_{\mathbf{k}_\gamma, \lambda}^* \hat{\mathbf{P}} | \varphi_{n_m, \mathbf{k}_i} \rangle \langle \langle 1 | \hat{a}_{\mathbf{k}_\gamma, \lambda}^\dagger | 0 \rangle \rangle$ stems from the interaction Hamiltonian (Eq. (6)), with the range of photon momenta being constraint by phase-matching (PM). Analogous to the two-level model of the main manuscript, this can be diagonalized to give the well-known eigenenergies

$$E_{\pm}(n_i, \mathbf{k}_i) = \frac{(\epsilon_{n_m, \mathbf{k}_i} - \epsilon_{n_i, \mathbf{k}_i}) + \omega_{\mathbf{k}_\gamma}}{2} \pm \sqrt{\frac{1}{4}((\epsilon_{n_m, \mathbf{k}_i} - \epsilon_{n_i, \mathbf{k}_i}) - \omega_{\mathbf{k}_\gamma})^2 + |V(n_i, \mathbf{k}_i)|^2}.$$

The corresponding eigenstates are obtained by the transformation

$$\hat{T}(n_i, \mathbf{k}_i) |\varphi_{n_m, \mathbf{k}_i} \rangle |0\rangle \geq \sum_{\pm} c_{e\pm}(n_i, \mathbf{k}_i) |\phi_{\pm}^{\text{pol}}(n_i, \mathbf{k}_i) \rangle \quad (13)$$

$$\text{with } c_{e\pm}(n_i, \mathbf{k}_i) = \left(1 + \frac{|V(n_i, \mathbf{k}_i)|^2}{(E_{\pm}(n_i, \mathbf{k}_i) - \omega_{\mathbf{k}_\gamma})^2} \right)^{-\frac{1}{2}}.$$

Thereby, the original matrix element transforms into

$$\begin{aligned} & \langle \varphi_{n_m, \mathbf{k}_i} | \langle 0 | \hat{U}(t, 0) | \varphi_{n_m, \mathbf{k}_i} \rangle | 0 \rangle \\ & = \langle \varphi_{n_m, \mathbf{k}_i} | \langle 0 | \hat{T}^\dagger(n_i, \mathbf{k}_i) \hat{T}(\dots) \hat{U}(t, 0) \hat{T}^\dagger(\dots) \hat{T}(\dots) | \varphi_{n_m, \mathbf{k}_i} \rangle | 0 \rangle \end{aligned} \quad (15)$$

$$= \left(\sum_{\pm} c_{e_{\pm}}^*(n_i, \mathbf{k}_i) \langle \phi_{\pm}^{\text{pol}}(\dots) | \right) \underbrace{\hat{T}(\dots) \hat{U}(t, 0) \hat{T}^{\dagger}(\dots)}_{e^{-i E_{\pm}(\dots) t}} \left(\sum_{\pm} c_{e_{\pm}} | \phi_{\pm}^{\text{pol}}(\dots) \rangle \right) \quad (16)$$

$$= \sum_{\pm} |c_{e_{\pm}}(n_i, \mathbf{k}_i)|^2 e^{-i E_{\pm}(n_i, \mathbf{k}_i) t} \quad (17)$$

For the overall dynamic structure factor near phase-matching (PM), this entails

$$S^{\text{pol}}(\mathbf{q}, \omega) = \frac{1}{2\pi} \int_{-\infty}^{\infty} dt e^{i\omega t} \int d^3x \int d^3x' e^{-i \cdot (\mathbf{x}-\mathbf{x}')} \langle \hat{n}(\mathbf{x}, t) \hat{n}(\mathbf{x}', 0) \rangle |_{PM} \quad (18)$$

$$= \frac{1}{2\pi} \int_{-\infty}^{\infty} dt e^{i\omega t} \int d^3x \int d^3x' e^{-i\mathbf{q} \cdot (\mathbf{x}-\mathbf{x}')} \frac{1}{V_0} \sum_{n_i}^{\text{VB}} \sum_{\mathbf{k}_i}^{\text{1.BZ}} \langle \varphi_{n_i, \mathbf{k}_i} | \hat{n}(\mathbf{x}) | \varphi_{n_m, \mathbf{k}_i} \rangle \times \sum_{\mathbf{k}_\gamma} \sum_{\pm} |c_{e_{\pm}}(n_i, \mathbf{k}_i, \mathbf{k}_\gamma)|^2 e^{-i E_{\pm}(\dots) t} \langle \varphi_{n_m, \mathbf{k}_i} | \hat{n}(\mathbf{x}') | \varphi_{n_i, \mathbf{k}_i} \rangle \quad (19)$$

$$= \frac{1}{V_0} \sum_{n_i}^{\text{VB}} \sum_{\mathbf{k}_i}^{\text{1.BZ}} \left(\left| \int d^3x e^{-i\mathbf{q}\mathbf{x}} \langle \varphi_{n_i, \mathbf{k}_i} | \hat{n}(\mathbf{x}) | \varphi_{n_m, \mathbf{k}_i} \rangle \right|^2 \sum_{\mathbf{k}_\gamma} \sum_{\pm} |c_{e_{\pm}}(\dots)|^2 \delta(E_{\pm}(\dots) - \omega) \right) \quad (20)$$

We find $S^{\text{pol}}(\mathbf{q}, \omega)$ to be the sum of partial scattering factors for all initial electronic states $\varphi_{n_i, \mathbf{k}_i}$ - each dressed by a two-level system, subject to energy conservation.

Evaluating the full expression for $S^{\text{pol}}(\mathbf{q}, \omega)$ numerically, would resemble a Floquet-type treatment of light-dressed electronic structure. However, such analysis is beyond the scope of our present experimental discussion of the EUV-polariton. Instead, we opt to interpret the summation in Eq. (20) as an average over all involved two-level systems resulting approximately in a single, effective TLS that dresses the regular dynamic structure factor known from IXS-measurements:

$$S^{\text{pol}}(\mathbf{q}, \omega) \approx S^{\text{IXS}}(\mathbf{q}, \omega) \cdot \sum_{\pm} |c_{e_{\pm}}|^2 e^{-(E_{\pm} - \omega)^2 / 2\Gamma_{\text{in}}^2} \quad (21)$$

To this end, we assume that any relative variation among electronic energies is small compared to the average transition energy ω_e . We apply the same reasoning to photonic modes, taking their frequency to be ω_γ on average. Small detunings either way are effectively accounted for by an intrinsic broadening factor Γ_{in} , while the overall coupling of electronic and photonic sectors is now mediated by the collective coupling V in

$$H_{\text{eff}}^{\text{pol}} = \hbar \begin{pmatrix} \omega_\gamma & 0 \\ 0 & \omega_e \end{pmatrix} + \begin{pmatrix} 0 & V \\ V^* & 0 \end{pmatrix}.$$

This reduced, effective model forms the basis of our experimental interpretation in the main manuscript (see its Eq. (1)), as it proves sufficient to explain all observed phenomena.

Despite the manuscript's experimental focus, we thank the reviewer for inspiring this broader theoretical discussion as well. Given that its content could also be of interest to other readers, we have included an abridged version of the above within the methods section (6) of our manuscript, accompanied by the reference:

p. 6 – l. 9: This approach corresponds to the low-excitation limit of a collectively-coupled system of emitters^{14,39,40} and captures the effective polaritonic response of the material (cf. Methods 6).

In addition, we have changed all previous occurrences of 'virtual' excitations in the manuscript into more appropriate formulations, i.e.,

- p. 5 – l. 15: [...] (virtual) electronic excitations [...]
- > [...] highly excited electronic states [...]
- p. 5 – l. 26: [...] a virtual electronic excitation [...]

- > [...] electronic excitations [...]
- p. 6 – l. 25: [...] virtual state [...]
- > [...] excited state [...]
- p. 13 – l. 26: [...] virtual photon's wavevector [...]
- > [...] idler photon's wavevector [...]

2) The emergence of the polariton is analysed in a phenomenological way, based on the simple two-level Rabi system (or, equivalently, the coupled-harmonic-oscillators model), without any damping rates included.

My first objection to this approach concerns exactly the damping, and the conclusion of having achieved strong coupling based on the width of the anticrossing and the linewidth of the *observed scattering yield*. But the true definition of strong coupling relies on the two systems (here photons and electrons) exchanging energy faster than they lose it to their own decay channels, meaning that the comparison should somehow be made between the width of the anticrossing, and the linewidths of the two individual coupled states---see for example the long discussion on coupling criteria in Torma & Barnes - Rep. Prog. Phys. 78, 013901 (2015).

We thank the reviewer for their critical comment on 'strong coupling' and understand that this terminology is subject to debate. However, we do not perceive a contradiction or incompatibility of our finding with the specific reference cited by the reviewer (viz. Törmä and Barnes). In fact, we have been well aware of this article, wherein the authors propose a broad definition of strong coupling. They state: 'For the purposes of this article, we define the term strong coupling in a pragmatic way: the system is in the strong coupling regime whenever the Rabi split is experimentally observable.' [40]. This condition is clearly met by our observation of the two-fold, split signature.

Other prominent authors in the field mirror this assessment in similar spirit: 'Note that strong coupling is measured not by the coupling magnitude but by the observability of its consequence' [15], or caution against strict adherence to a particular criterion: '[...] the strong-coupling regime is entered when their mutual interaction overcomes decoherence in the system, i.e., $g \gtrsim \gamma$, where γ is a typical decoherence rate (the exact criterion that should be used is somewhat arbitrary with several valid choices [...])' [12].

Similar to the reviewer, the last reference rightfully emphasizes the necessity of overcoming decoherence (i.e., decay) for strong coupling phenomena to emerge. However, its authors do not focus on individual decay channels, but refer more broadly to 'typical decoherence'. In the most general (and also most restrictive) sense, we read this as the overall decay rate or broadening.

Adhering to this criterion, we find our approach once again compatible with Törmä and Barnes [40]. In their discussion on the influence of damping (Sec. 3.2 [3]), they state the strong-coupling condition

$$A > \gamma^2/2 + \gamma_{\text{SPP}}^2/2.$$

Therein, A is the square of the Rabi-frequency (or $\sqrt{A} = 2V$, as far as our model is concerned), while γ and γ_{SPP} mark the individual damping rates in their setting. Notably, they do not include further instrumental broadening. For our model, we have opted to summarize all individual damping rates as well as instrumental-broadening into a single rate $\Gamma^2 = \Gamma_e^2 + \Gamma_\gamma^2 + \Gamma_{\text{inst.}}^2 > \Gamma_e^2 + \Gamma_\gamma^2$, which is naturally broader than the individual channels. We can then formulate an analogous strong-coupling condition:

$$2V > \Gamma = \sqrt{(\Gamma_e^2 + \Gamma_\gamma^2 + \Gamma_{\text{inst.}}^2)},$$

which effectively contains *all* damping rates encountered in our system. Notably, this renders our condition even stricter than necessary in the ideal case of $\Gamma_{\text{inst.}} = 0$.

In this sense, we maintain that our model would not benefit from further dissemination of the damping rates, as they are anyways convoluted in the overall broadening. At the same time, our current model already provides indications of strong coupling, even though it is overly restrictive on Γ .

This notwithstanding, we fully agree with the reviewer that considerations on strong coupling should ideally be formulated cautiously – aware of the pertinent discussions. For the respective indications seen in our experiment, we thus suggest a couple of re-formulations in the manuscript:

- p. 1 – l. 29: This insight opens a new pathway to explore strong-coupling [...]
→ This insight could open a new pathway to explore strong-coupling [...]
p. 2 – l. 7: [...] we corroborate our hypothesis of EUV-strong-coupling and [...]
→ [...] obtain further indication of EUV-strong-coupling and [...]
p. 5 – l. 15: [...] it strongly couples [...]
→ [...] it couples [...]

Moreover, we shall include the suggested reference of Törmä and Barnes [40] alongside the other new references for broader context on the subject of strong coupling in a revised paragraph:

- p. 8 – l. 12: Having thus obtained a sensitive probe of the anti-crossing region (Fig. 4 d), we focus again on the level splitting found there (zoomed inset) and put its magnitude into perspective: At $2V = 1.64$ eV, the Rabi splitting is comparable to the overall broadening $\hbar\Gamma = 1.64$ eV. This condition ($2V \gtrsim \hbar\Gamma$) – alongside the visibility of the splitting itself – is conventionally considered to mark the onset of the coupling regime^{5,6,12,14,15,40}. As such, our results indicate that the EUV-polariton could reach strong-coupling conditions intrinsically. This is particularly remarkable in contrast to more typical strong-coupling schemes, which require cavity-enhancement⁴⁻⁷, and thus encourages to further explore the underlying coupling mechanism.

A second objection concerns the description based on a single two-level system. Based on the sketch of Fig. 3a, one would anticipate that it is several virtual transitions that participate simultaneously in the coupling. This, then, should be either described with a Dicke-like model [Dicke - Phys. Rev. 93, 99 (1954)] in the case of fermions, or with a bosonic description in the case where all these states form a collective dipole moment together [see e.g. Tserkezis et al. - Rep. Prog. Phys. 83, 082401 (2020)].

Conceptually, we fully agree with the reviewer: The collective interaction of multiple emitters or electronic transitions, for that matter, underpins the polaritonic effect that we observe. This scenario can be captured via the overall electronic structure of the solid (as in our theoretical motivation above) or represented via a set of ‘atomic’ emitters – as in a Dicke-like model. However, we do not require the full complexity of either approach to describe our observed polaritonic phenomenology. In particular, we do not require the high number of potential excitations that a Dicke or Tavis-Cummings model could accommodate. Instead we can focus on the case of a single excitation at a time, because we use the comparatively rare XPDC process to excite the EUV-polariton. Doing so will lead us back to the much simpler description in terms of a single two-level system, as we outline below.

We consider the Hamiltonian for N atomic emitters coupled via a photonic mode

$$\hat{H} = \hbar\omega_\gamma \hat{a}^\dagger \hat{a} + \frac{\hbar\omega_e}{2} \sum_{j=1}^N \sigma_j^z + V_{atom} \sum_{j=1}^N (\sigma_j^+ \hat{a} + \text{h.c.}) - E_0$$

Here, σ are Pauli matrices per atom, $\sigma^+ = 1/2(\sigma^x + i\sigma^y)$, \hat{a} and \hat{a}^\dagger are photonic ladder operators and $E_0 = -N\hbar\omega_e/2$ is the ground state energy – subtracted for convenience. If we limit this model to low excitation numbers afforded by the XPDC process (i.e., 1 or 0 photons). The corresponding states are:

$$|\phi_\gamma\rangle = |\downarrow \dots \downarrow\rangle_{atoms} \otimes |1\rangle_{photons}$$

$$|\phi_e\rangle = \frac{1}{\sqrt{N}} \sum_{j=1}^N |\downarrow \dots \uparrow_j \dots \downarrow\rangle_{atoms} \otimes |0\rangle_{photo}$$

Within the rotating-wave approximation, these two levels form a closed subspace and their projected Hamiltonian becomes H^{pol} of the manuscript (Eq. (1)), with the new collective coupling strength $V = \sqrt{N} V_{\text{atom}}$ (similar reasoning can be found, e.g., in Refs. [14,39,40]). For the sake of simplicity, we have adopted the resulting, effective TLS-model from the beginning of our manuscript. In order to alert readers to the more fundamental connection, we have added the following remark in the manuscript:

p. 6 – ll. 8: This approach corresponds to the low-excitation limit of a collectively-coupled system of emitters³⁹ and captures the effective polaritonic response of the material (cf. Methods 6).

3) Similarly, if the polaritonic description is to be adopted, the authors should provide some guidelines for calculating (at a first stage in a simplified picture) the coupling strength V —which again relates to specifying the actual nature of the transition that takes place.

We appreciate the reviewer’s interest in obtaining a prescription for calculating the coupling strength V . This aspiration aligns fully with our own long-term goal to obtain a comprehensive theoretical description of the EUV-polariton and, ultimately, achieve predictive capabilities. For the moment, however, our focus is chiefly on reporting the experimental observation of the modulated XPDC signal cone along with its polaritonic interpretation. We emphasize that our phenomenological two-level model already conveys this qualitatively new insight, whereas truly quantitative developments will present distinct steps beyond the current scope of discussion.

Without having a full, first-principles’ theory established, we can nevertheless outline an approximate assessment of V – based on our earlier *Theoretical motivation of the TLS-model*. In doing so, we follow the reviewer’s suggestion to stay ‘in a simplified picture’, from which we can already obtain a valuable cross check regarding the plausibility of our model’s coupling parameter.

As our starting point, we shall try to compute the expectation value of the coupling: Its linear order $\langle \hat{V} \rangle$ will vanish, given that interaction Hamiltonian is anti-diagonal (cf. Eq. (12)), while its square $V^2 = \langle \hat{V}^\dagger \hat{V} \rangle$ has a non-trivial expectation value. Inserting the individual matrix elements $V(n_i, \mathbf{k}_i)$ and summing over all possible initial states, this gives:

$$\begin{aligned} V^2 &= \frac{1}{V_0} \sum_{n_i}^{\text{VB}} \sum_{\mathbf{k}_i}^{\text{1.BZ}} V^\dagger(n_i, \mathbf{k}_i) V(n_i, \mathbf{k}_i) \\ &= \frac{1}{V_0} \sum_{n_i}^{\text{VB}} \sum_{\mathbf{k}_i}^{\text{1.BZ}} \sum_{\mathbf{k}_\gamma}^{\text{PM}} \sum_{\lambda} \frac{2\pi}{V\omega_{\mathbf{k}_\gamma}} \left| \langle \varphi_{n_i, \mathbf{k}_i} | \epsilon_{\mathbf{k}_\gamma, \lambda}^* \hat{\mathbf{p}} | \varphi_{n_m, \mathbf{k}_i} \rangle \right|^2 \underbrace{\langle 0 | \hat{a}_{\mathbf{k}_\gamma, \lambda} \hat{a}_{\mathbf{k}_\gamma, \lambda}^\dagger | 0 \rangle}_1 \end{aligned}$$

Here, we have already made use of the fact that creation and annihilation of the idler photon have to affect the same mode \mathbf{k}_γ pairwise. Next, we will constrain the modes further by making the phase-matching condition (PM) explicit, i.e., enforcing Eq. (M4) using a delta-function:

$$\begin{aligned} V^2 &= \frac{1}{V_0} \sum_{n_i}^{\text{VB}} \sum_{\mathbf{k}_i}^{\text{1.BZ}} \sum_{\mathbf{k}_\gamma} \sum_{\lambda} \frac{2\pi}{V\omega_{\mathbf{k}_\gamma}} \left| \langle \varphi_{n_i, \mathbf{k}_i} | \epsilon_{\mathbf{k}_\gamma, \lambda}^* \hat{\mathbf{p}} | \varphi_{n_m, \mathbf{k}_i} \rangle \right|^2 (2\pi)^3 \delta(\mathbf{k}_\gamma - (\mathbf{k}_p - \mathbf{k}_s + \mathbf{G})) \\ &= \frac{1}{V_0} \sum_{n_i}^{\text{VB}} \sum_{\mathbf{k}_i}^{\text{1.BZ}} \sum_{\lambda} \int d^3k_\gamma \frac{2\pi}{\omega_{\mathbf{k}_\gamma}} \left| \langle \varphi_{n_i, \mathbf{k}_i} | \epsilon_{\mathbf{k}_\gamma, \lambda}^* \hat{\mathbf{p}} | \varphi_{n_m, \mathbf{k}_i} \rangle \right|^2 \delta(\mathbf{k}_\gamma - (\mathbf{k}_p - \mathbf{k}_s + \mathbf{G})) \\ &= \frac{1}{V_0} \sum_{n_i}^{\text{VB}} \sum_{\mathbf{k}_i}^{\text{1.BZ}} \sum_{\lambda} \frac{2\pi}{\omega_{\mathbf{k}_\gamma}} \left| \langle \varphi_{n_i, \mathbf{k}_i} | \epsilon_{\mathbf{k}_\gamma, \lambda}^* \hat{\mathbf{p}} | \varphi_{n_m, \mathbf{k}_i} \rangle \right|^2 \end{aligned}$$

Having converted the mode summation into an integral, the delta-function fixes $\mathbf{k}_\gamma = (\mathbf{k}_p - \mathbf{k}_s + \mathbf{G})$. Henceforth, the symbol \mathbf{k}_γ is to be understood merely as a fixed shorthand for the sum of scattering momenta. In order to evaluate (or approximate) the remainder of the above expression, we can take inspiration from the dielectric function $\epsilon_2(\omega)$ – written in its Kubo-Greenwood form (cf. [C1]):

$$\varepsilon_2(\omega) = \frac{8\pi^2}{\omega^2} \frac{1}{V_0} \sum_{n_i}^{\text{occ}} \sum_{n_f}^{\text{unocc}} \sum_{\mathbf{k}}^{\text{1.BZ}} \left| \langle \varphi_{n_i, \mathbf{k}} | \hat{\epsilon} \hat{\mathbf{p}} | \varphi_{n_f, \mathbf{k}} \rangle \right|^2 \delta(E_{n_f, \mathbf{k}} - E_{n_i, \mathbf{k}} - \omega)$$

For consistency, we first have to treat this to the same approximations that we have imposed on our band structure model earlier: We limit the summation in n_i to valence-bands (VB) and posit that an excitation by $\sim 100\text{eV}$ will match most closely with exactly one final band n_f for each \mathbf{k} -point within the overall bandwidth Γ (cf., two-level approximation above). For the selected transition, we consider the energetic delta-condition to be satisfied within this bandwidth Γ – convoluting a normalized Gaussian. The resulting expression for $\varepsilon_2(\omega)$ in the TLS-band-structure reads:

$$\varepsilon_2(\omega) \approx \frac{8\pi^2}{\omega^2} \frac{1}{V_0} \sum_{n_i}^{\text{VB}} \sum_{\mathbf{k}_i}^{\text{1.BZ}} \left| \langle \varphi_{n_i, \mathbf{k}_i} | \hat{\epsilon} \hat{\mathbf{p}} | \varphi_{n_f, \mathbf{k}} \rangle \right|^2 \frac{1}{\sqrt{2\pi}\Gamma}$$

Inserting this into the expectation value of V^2 and summing equally over polarizations (λ), we find:

$$V^2 \approx \frac{1}{V_0} \sum_{n_i}^{\text{VB}} \sum_{\mathbf{k}_i}^{\text{1.BZ}} \frac{4\pi}{\omega_{\mathbf{k}_\gamma}} \left| \langle \varphi_{n_i, \mathbf{k}_i} | \hat{\epsilon}_{\mathbf{k}_\gamma, \lambda}^* \hat{\mathbf{p}} | \varphi_{n_m, \mathbf{k}_i} \rangle \right|^2 = \varepsilon_2(\omega_{\mathbf{k}_\gamma}) \frac{\omega_{\mathbf{k}_\gamma} \Gamma}{\sqrt{2\pi}}$$

Finally, we need to insert \hbar^2 on the right-hand side to convert back from atomic units and can evaluate the expression numerically: We obtain the Kubo-Greenwood-type dielectric function from DFT simulations of diamond (here, using the FHI-aims code [C2] and the HSE06 exchange potential) - yielding $\varepsilon_2(100\text{eV}) \approx 0.008$. Combined with $\hbar\omega_{\mathbf{k}_\gamma} = 100\text{eV}$ and $\hbar\Gamma = 1.64\text{eV}$, we compute the overall value of the coupling strength to be $V \approx 0.72\text{ eV}$.

This result is remarkably close to the value, which we have obtained from fitting the measured data with our TLS-model (i.e., $V_{\text{fit}} = 0.82\text{ eV}$) – especially when considering the coarse nature of the presented approximation. Reaching such robust agreement by comparatively simple means, we hope to convince the reviewer that our modelling approach is indeed plausible. At the same time, we want to re-iterate the important caveat that the calculations above give merely an estimate and cannot stand in for a comprehensive theory of the EUV-polariton. The latter remains an essential goal for our future developments.

4) In any case, the narrowness of the splitting, and the shape of the data in Fig. 4c, suggest something closer to a Fano resonance as a result of this narrow transition (let's assume it is indeed a virtual electronic transition) over a wider photonic background. I wonder if the data of Fig. 4c would not be better reproduced by the Fano formula [Fano - Phys. Rev. 124, 1866 (1961)].

The reviewer is right in their observation that our measured line-shapes could also be reproduced by a Fano formula. However, we want to stress that such an analysis would be just as phenomenological – if not more speculative – than our polaritonic model. In fact, earlier studies on x-ray parametric down-conversion by Tamasaku et al. were fitted by the Fano-formula, suggesting a hypothetical interference effect between Compton scattering (\rightarrow ‘wider photonic background’) and XPDC (\rightarrow ‘narrow transition’). However, these dissimilar final states should not interfere quantum mechanically. Tamasaku et al., themselves caution their readers to this effect, stating: ‘One of the most controversial points of the interpretation is the fact that the final state of the parametric down-conversion might be different from that of the Compton scattering even when absorption of the idler wave is taken into account, so that the interference effect could not be expected.’ [4] Our polaritonic interpretation, actually, resolves this conundrum by reconciling the disparate states into common, hybridized precursor states (see also manuscript p. 5).

We have added this aspect about the Fano-formula in the Methods section 7: Fitting procedure of our manuscript.

p. 15 - ll. 37: We note that the line shape could also be fitted by the Fano-formula [48] on a phenomenological basis. This was done by Tamasaku et al. under similar conditions, suggesting an interference between Compton scattering and XPDC [35]. However, these processes should not interfere quantum mechanically (see also Tamasaku et al. [49]), which renders a Fano-model inapplicable. In contrast, our polaritonic interpretation reconciles both processes and yields Eq. 12 (Methods) to describe the full, hybridized phenomenon.

As a couple of suggestions for further work that could shed light on these issues, I wonder if it would be possible to perform the same experiment with a different material, other than diamond, or with a different crystallographic facet of the same sample, so as to pick a different reciprocal-lattice vector, and thus generate different signal/idler photons. In a similar direction, what would happen with larger rotations of the same sample?

We thank the reviewer for their keen interest in the new effect and for their suggestions on further explorations. These are very reasonable directions. It is certainly possible to detect the effect in other materials and we are indeed planning to investigate crystals such as silicon and gallium-arsenide in future experimental campaigns. Exploratory measurements to this end have been very promising, yet require more synchrotron beamtime and analysis.

Concerning different reciprocal lattice-vectors, we have acquired multiple data sets in diamond – at excitations both on and off resonances (i.e., using different combinations of signal/idler energies). All show polaritonic signatures that are compatible with our model, while their mapping to structural information (cf. QED-based approach [38]) is still work in progress.

Finally, for larger sample rotations than shown in the manuscript, the phase-matching condition for XPDC is no longer fulfilled. There, we have observed no signal cone – as expected. Actually, we use this fact to acquire suitable background images.

Overall, I believe that the experiment itself could justify publication in Nature Communications, but if the authors want to insist on the picture of the UV polariton, more evidence and work is needed.

We are very grateful for the reviewer's positive assessment of our experiment on its own and their suggestion towards publication of the results in Nature Communications. Moreover, we appreciate the reviewer's detailed and constructive feedback on our polaritonic interpretation. We hope that we could address their concerns in the above discussion and convince them that there is additional merit to the model, which warrants its publication as well. Even if the reviewer should remain more skeptical of our polaritonic interpretation than the other two reviewers were, we want to point out that this skepticism – and its resulting, constructive discussion – may be the best reason for publication. As such, the broad readership of Nature Communications can participate in the dissemination of our interpretation and accelerate the iteration towards more comprehensive theories.

Reviewer #2 (Remarks to the Author):

The article "X-ray parametric down-conversion reveals EUV-polariton" is a noteworthy work in the field of X-ray parametric down conversion. The significance/advancement is mainly regarding the identification of the EUV-polariton mechanism. This is interesting to researchers in the field, as the experimental evidence presented herein helps solidify the mechanism of the X-ray PDC and, as the authors note, opens up new research avenues. Combining the experimental results with the TLS model bolsters the claim and helps to support the conclusions of the paper. I do not find any flaws in the analysis, interpretation, or conclusions, and the methodology seems sound. My overall recommendation is acceptance with some relatively minor revisions.

We are very grateful for the reviewer's thoroughly positive assessment of our work and their recommendation to publish in Nature Communications. Below, we will address the minor revisions suggested by the reviewer.

Regarding claims of resolving the XPDC cone for the first time, e.g. on:

-> Lines (25)-(26): "Here, we resolve the full down-conversion cone at x-ray 25 wavelengths for the first time [...]"

-> Line (47): "This enables us to resolve the conversion cone of XPDC for the first time [...]"

-> Line (59)-(60): "Remarkably, a full signal cone has never before been observed in the x-ray regime [...]"

-> Line (90)-(91): "The introduction of 2D imaging fundamentally distinguishes our setup from all previous approaches to measure XPDC [...]" (and other places)

I do not believe this work constitutes the first spatially resolved/2D imaging of cone measurements of the XPDC process. I would draw the author's attention to arXiv:2310.13078, recent work from the NSLS-II, which also claims to resolve the down conversion cone with an area detector for the first time. I would encourage the authors to consider/cite their work and either (1) remove claims of the 'first time' etc. or, if they feel the referenced work does not fully meet the claim but their work does (2) address this point explicitly.

We thank the reviewer for drawing our attention to the recent manuscript by Goodrich et al. [25] – dating to 19th October 2023 – which presents very nice results on the detection of XPDC in the degenerate case (i.e., photon energies $w_s = w_i$). From the cited manuscript, we cannot assess when their measurements were conducted. However, inference from the lead-author's presentation on the subject, which we attended at the SRI2024 conference, suggested an observation in the year 2023. Just for the record, we note that our presented measurements on non-degenerate XPDC were performed during two campaigns at the ESRF (HC-4489 and HC-4907), which took place from 28th April – 5th May 2020 and 1st March - 7th March 2021, respectively.

Notwithstanding the date of observation, we think that both measurements constitute valuable achievements in their own right. In particular, the two approaches are so distinct in their setup, method and ultimate aim that any further distinction based on their timing should be subordinate. We will gladly resolve any ambiguity and avoid potential interference by specifying the (highly) non-degenerate case of XPDC observed in our measurements. In this spirit, we address the text-passages that the reviewer has remarked on as follows:

p. 1 – ll. 25 – 26: Here, we resolve the full down-conversion cone at x-ray wavelengths for the first time [...]

→ Here, we resolve the full signal cone of non-degenerate down-conversion at x-ray wavelengths for the first time [...]

p. 2 – l. 5: This enables us to resolve the conversion cone of XPDC for the first time [...]

→ This enables us to resolve the signal cone of non-degenerate XPDC for the first time [...]

p. 2 – ll. 19: Remarkably, a full signal cone has never before been observed in the x-ray regime, whereas its equivalents from visible PDC are well known (e.g., Ref.21).

→ While this cone-shaped emission is well known for visible PDC (e.g., Ref.²⁴), it could only recently be observed for x-rays in the degenerate²⁵ and – here – the non-degenerate case of XPDC.

This point also lends a good opportunity to reference the respective results:

p. 3 – ll. 17: The introduction of 2D imaging fundamentally distinguishes our setup from all

previous approaches to measure XPDC (e.g., Refs.26,27,28,29,30-34).
→ The introduction of 2D detection in combination with an imaging analyzer distinguishes our setup fundamentally from all previous approaches to measure XPDC (e.g., Refs.^{23,30-36}).

In addition, we have identified two further passages, which we clarify in similar fashion:

p. 2 – l. 23: [...] to resolve the signal cone for the first time

→ [...] to resolve the signal cone both spatially and spectrally

p. 8 – l. 21: We reported on the first observation of a full PDC cone in the x-ray regime [...]

→ We reported on the first observation of a full signal cone for non-degenerate PDC in the x-ray regime [...]

On a related note, the authors claim (emphasis mine) to "resolve the **full** down-conversion" in the abstract, but then later on lines (67-68) specify: "We focus specifically on down-conversion into signal photons [...] while the idler remains unobserved". I find the language of saying the "'full' down-conversion is resolved" at odds with the fact that the lower energy photon is not observed in these measurements. I would recommend rewriting/clarification.

We thank the reviewer for identifying the formulation '[...] resolve the full down-conversion cone [...]' as a source for potential misunderstanding. Even though we implied a singular cone, we agree that it could be misread for the full effect – especially by readers who are mostly familiar with the superimposed cones of the degenerate PDC-case. We have already addressed the respective formulation within one of the above revisions - now, specifying explicitly the 'signal cone'. In conjunction with this re-formulation, we also seek to emphasize the distinction between signal and idler more strongly in the setup part by expanding:

p. 2 – ll. 27: We focus specifically on down-conversion into signal photons at $\hbar\omega = 9.69$ keV and idler photons at $\hbar\omega = 100$ eV [...]. While the idler remains unobserved, we filter for the emitted signal using one of the spectrometer's spherically-bent crystal analyzers (SBCA).

→ We focus specifically on non-degenerate (i.e., asymmetric) down-conversion into signal photons at $\hbar\omega = 9.69$ keV and idler photons at $\hbar\omega = 100$ eV [...]. We filter specifically for the signal photon using one of the spectrometer's spherically-bent crystal analyzers (SBCA), while the idler remains unobserved, due to its interaction with the material.

We have further checked the remainder of the manuscript for ambiguous passages, making sure that we report on fully resolving the signal cone and not the entire effect. Our previous revisions had covered all but the following instance already:

p. 3 – l. 19: [...] capture the entire XPDC cone [...]

→ [...] capture the entire XPDC signal cone [...]

We hope that the respective changes help to clarify our statement.

These issues notwithstanding, the work presented is of high quality, interesting to researchers in the field, and is recommended for publication.

We thank Reviewer #2 for their constructive and concrete feedback, which was of immediate help to improve the manuscript. We hope that the resulting changes align with their intentions. Moreover, we want to express our gratitude once again for the reviewer's positive evaluation of our work and endorsement of its publication in Nature Communications.

Reviewer #3 (Remarks to the Author):

The development and implementation of non-linear techniques at higher photon energies is an important area of intense research. One approach is XPDC. In the present manuscript, the authors report a remarkable progress in this direction. They observe the full XPDC cone at once for the first time, explore its shape across the entire XPDC ellipsoid, and identify the formation of an EUV polariton. Strikingly, the analysis reveals that it is naturally close to the strong-coupling limit. The comprehensive experimental data is convincingly supported by adequate theory modelling. These developments are of high and broad significance, and open a clear path to a broad range of interesting follow-up works, as hinted at in the discussion. In my view, these results fully merit a publication in Nature Communications.

We thank the reviewer for their very positive feedback to our manuscript and the recommendation for its publication in Nature Communications. In the following, we will address the reviewer's questions on a point-by-point basis.

Before publication, I have few questions:

* Is the "universality of polaritons" stressed in the introduction really an important aspect of the work? Theoretically, it is not very surprising?

We fully agree conceptually, but disagree on the practical implications. Theoretically, the universality of polaritons is indeed not surprising. For any wavelength regime of light that can be matched with a commensurate excitation in matter, there should exist polaritonic hybrid states. Following this general paradigm, also the existence of EUV-polaritons could have been inferred a priori. Nevertheless, EUV-polaritons were neither expected in XPDC nor previously identified as such – possibly because polaritons are more readily associated with their manifestations at lower energies.

Addressing this perception, we wanted to emphasize the universality of polaritons as early as possible in the introduction and, thus, put our finding of the EUV-polariton into this broader context. We have rephrased our introduction to make this intention clearer as follows:

p. 1 – ll. 39: At shorter wavelength, however, polaritonic phenomena remain largely unexplored. In this work, we present evidence for polariton formation in the EUV, simultaneously highlighting the universality of light-matter hybridization and opening up a wholly new regime for its investigation.

→ At shorter wavelength, analogous polaritonic hybridization stands to be expected, given the universal nature of QED. However, the lack of suitably reflecting cavities has so far precluded the realization of strong-coupling conditions under the conventional, cavity-based paradigm – leaving polaritonic phenomena largely unexplored in the EUV and x-ray range.

In this work, we present evidence for polariton formation in the EUV via a different route and thus open up a new regime of light-matter hybridization for investigation.

* What is the overall conversion efficiency of incident photon into XPCS photon (created / detected)? What is the loss due to the energy analysis / imaging? How does the overall efficiency compare to the previous experiments, given that the authors claim superior data-acquisition efficiency?

We appreciate the reviewer's questions on efficiency and want to take this opportunity to clarify potential ambiguities:

First of all, it is important to separate physical 'conversion efficiency' – which could be compared across experiments under certain circumstances – from the practical 'efficiency in data acquisition'. Where we remark on the latter (as referenced in the reviewer's last question), we want to emphasize the ease of use that our new 2D detection scheme offers compared to raster-scanning approaches under the same conditions. However, we do not intend any blanket claim of superiority. In particular, our comment is also a reflection on our own previous approaches to scan XPDC phase-matching conditions using high-resolution x-ray diffraction setups with very constrained angular acceptance (see

Ref. [23]). We have extended our remark in the manuscript to this end as follows:

p.3 – ll. 18: It improves our efficiency in data-acquisition substantially, as we can capture the entire XPDC cone in a single frame – effectively enabling us to trace out the phase matching condition with unprecedented resolution.

→ It improves our efficiency in data-acquisition substantially, as we can capture the entire XPDC signal cone in a single frame, rather than resolving it by point-wise raster scan (cf., e.g., Ref. [23]). This, effectively, enables us to map out the phase-matching condition with unprecedented resolution.

For the sake of illustration, we could quantify the respective improvement by comparing the large angular field of view in the 2D case ($\sim 2 \text{ deg} \times 2 \text{ deg}$) with the stepwise angular acceptance of our setup in Ref. [23] (i.e., $57 \text{ mdeg} \times 1 \text{ mdeg}$). Arguably, this would require about $35 * 2000$ individual acquisitions to raster the same image for the same amount of counts and thus increase the acquisition time proportionally by 70000. However, such quantitative comparison hinges on the choice of parameters, ultimately, whereas we want to focus our emphasis on the qualitative novelty and ease of use, here.

Next, we turn our attention to the physical ‘conversion efficiency’ of XPDC, which the reviewer inquired about initially. On this subject, it is likewise crucial to avoid ambiguities: While a measure for ‘efficiency’ in terms of the simple ratio of outgoing photons (or photon pairs) over incident photons may lend itself to compare equivalent experiments, it will prove inadequate once setups differ substantially. In the latter case, such efficiency ratios would primarily reflect experimental choices (e.g., chosen bandwidths, captured solid angles, etc.), rather than providing information about the underlying effect. The reviewer themselves has already hinted at this problem, when mentioning ‘the loss due to the energy analysis’ in our case. We seek to avoid confusion over such potential ambiguities by reporting only the bare count rates and specifications of our setup – in line with several other XPDC reports [31,33,35] – but refrain from stating an efficiency ratio in our manuscript.

Nonetheless, we gladly put our experiment in context with other measurements for the reviewer: Computing the conversion efficiency from the peak count rate of signal photons ($\sim 0.5 \text{ ph/s}$ per pixel) generated per incident photon flux on sample ($1e13 \text{ ph/s}$), we find a ratio of $5e-14$ per pixel. If we instead choose to integrate the peak-to-dip signal across the complete detected image ($\sim 300 \text{ ph/s}$), the corresponding ratio gives a total efficiency of $3e-11$. We emphasize again that these values are specific to the presented setup at an incident photon energy of 9.79 keV , an incident bandwidth of approx. 1 eV and a detection bandwidth of the spherical analyzer of approx. 100 meV .

Assessing previous studies by the same measure, we can compute conversion efficiencies – first for the degenerate case of XPDC of:

$1.25e-11 = 1 \text{ (count/h)} / 2e7 \text{ incident photons/s}$ for Ref. [33] and

$6.7e-13 = 6 \text{ (counts/h)} / 2.5e9 \text{ incident photons/s}$ for Ref. [31].

Even though these ratios seem superficially similar to our result, it is important to observe that the reported counts were obtained by integrating across much larger detection-bandwidths (on the order of several keV). This also allows for a larger range of phase-matching conditions to be fulfilled at any given sample angle. Conversely, the underlying *generation efficiency* for two symmetrically down-converted photons per energy and angle interval in these cases must have been 4 to 5 orders of magnitude weaker.

For non-degenerate cases of XPDC, the study that resembles our scenario most closely was conducted by Tamasaku et al. [35]. Rocking across the outer edge of the phase-matching ellipse, they measure a spatially integrated peak-to-dip signal of 22 ph/s at an incident flux of $\sim 1e12 \text{ ph/s}$. This corresponds to an efficiency ratio of $2e-11$, which is again superficially comparable to our finding of $3e-11$. At this point, however, we should likewise take ‘the loss due to the energy analysis’ into account as the reviewer remarked. The result in Ref. [35] was obtained within a detection bandwidth of $\sim 2 \text{ eV}$, whereas our setup applies a much more narrow energetic filter of $\sim 100 \text{ meV}$, corresponding to a factor of ~ 20 less transmission. The fact that we still arrive at similar ‘efficiency’ ratios, irrespective of

this stark difference, underscores the limitations of the simplified measure. Here for instance, we are failing to account for detailed rocking-curve dependencies, angular distributions of the signal and potential cancellations if positive and negative contributions are smeared out. We conclude that the most dependable/reliable form of reporting our result is in terms of bare count rates – for readers to super-impose their own measures and comparisons as they see fit.

* Is Bragg scattering an issue? How does it compare to the Compton scattering background? Or is it fully removed by the analyzer?

Fortunately, Bragg scattering is not an issue in the presented case. Here, our detection energy $w_s = 9.69$ keV, i.e., the energy for which the analyzer is set to reflect efficiently, is detuned by 100 eV from the incident energy ($w_p = 9.79$ keV). Correspondingly, any Bragg reflection (or other elastic scattering) of the pump beam is strongly suppressed by several orders of magnitude (see Figure 2).

Fig. 2 - Calculated reflectivity curve of the Si(660) analyzer. The green line indicates the set point of the analyzer at $w_s = 9.69$ keV (using a backscattering angle of $\theta_b = 88.306$ deg), while the blue line indicates the detuned pump energy $w_p = 9.79$ keV.

Compton-scattering, however, cannot be filtered out by the analyzer as easily. There remains a background contribution of approx. 4 cnts/s per pixel. For our analysis, this is removed by recording a background image close to - but off - the phase-matching condition (i.e., where no XPDC cone is observed) and subtracting this image from all measurements on phase-matching conditions.

* Throughout the analysis, the authors claim "Confirming this pattern will allow us to unambiguously identify XPDC without coincident detection of any idler photon". It seems perfectly convincing, and I do not argue against it - in particular, given the good agreement with the later theory analysis. But is there actually a theoretical support or even a proof behind this statement?

We appreciate the interest of the reviewer in the characteristic signatures of XPDC and would like to address the question on their theoretical support in two stages:

First and fundamentally, the signature of XPDC can be determined from conservation of energy ($w_p = w_s + w_i$) and momentum ($k_p + G = k_s + k_i$) without the need for additional theory. This phase-matching condition fully constrains the kinematics of the process, if the usual photonic dispersion $w_j = c |k_j|$ is taken into account as well. As a result, XPDC is limited to the scattering ellipsoid shown in Fig. 2 (d) of the manuscript, which we trace with our measurements.

For a second level of identification, we can look to the QED-based framework that we have previously developed for the fully perturbative description of XPDC [23,38]. This predicts that each scattering cone is further modulated, due to the polarization of the idler photon (see also the discussion on the 'horseshoe'-structure below), which we likewise observe.

We have made the following additions to the manuscript, in order to clarify the fundamental nature of the phase-matching signature:

p. 2 – l. 15: [...] conversion process will only occur if its phase-matching condition is satisfied

→ [...] conversion process will only occur if its phase-matching condition is satisfied, which is fundamentally determined by energy and momentum conservation.

p. 2 – ll. 19: These cones constitute the characteristic scattering signature of (x-ray) PDC²³.

* Which part ("radius"/ opening angle) of the polariton structure is chosen to compare with the theoretical XPCS cone opening angles?

We have chosen the ‘flipping point’, i.e., the radius at which we observe a change of sign in the signal from positive to negative, as the reference to compare with theoretical opening angles. For Fig. 2 d, we have indicated this approach on

p. 4 – l. 25: In all slices, the change of sign aligns with the phase-matching ellipsoid of Fig. 1 d (reproduced as a green wire-frame in Fig. 2 d).

To make this particularly clear, we have added on

p. 4 – l. 26: Fitting the circular contrast, we extract the cones’ opening angles [...]

-> Fitting the circular contrast at this change of sign, we extract the cones’ opening angles [...]

* Fig 2(c) shows a horseshoe-structure of reduced counts, rather than the double-lobes in (b/c). Can this be understood from the polarization argument?

Indeed, both the ‘horseshoe-structure’ and the ‘double-lobes’ can be understood when considering the polarization properties of the EUV-idler quant. In the central part of figure 3, we illustrate how our theoretical model would predict the scattering patterns that correspond to the original Fig. 2 (a) - (c), both with (centre-left) and without (centre-right) the polarization dependence of Eq. 7 (Methods 5). While the full model captures the non-trivial structures around the cone well, the non-polarized version simplifies to full rotational symmetry.

Fig. 3 - Juxtaposition of measured XPDC signal cones at different rocking angles (left column; a-c) with respective simulations that include (centre-left; d-f) or disregard (centre-right; g-i) polarization effects of the EUV-idler (cf. Methods 5). The associated phase-matching conditions are sketched in the right-most column (j-l).

The physical origin of the pattern can be traced to the transverse polarization of the EUV-polariton and the way it is probed by XPDC: Accessing a specific reciprocal component (G) of the excitation, we are sensitive predominantly to the EUV-quant being polarized in that same direction. For the side-lobes (i.e., phasematching k_i out of plane), there is always a component of the polarization along G to be imaged. However, at the upper and lower point of the cone, the polariton's propagation direction (k_i) may align with G , rendering its polarization largely orthogonal. This is the case for the condition shown in Fig 2 (b), where no signal can be found around $\chi = 0$. Similarly, for the conditions shown in the figure's rows (a) or (c), k_i aligns with G at the lower- or uppermost phase-matching point, respectively. Again, this leads to signal suppression - while on each opposite side of the cone, the EUV-idler's polarization is allowed to be parallel to G , giving rise to strong signal and an overall 'horseshoe'-shape.

We have added this more detailed discussion of the polarization imprint to the Methods section 5 of the manuscript and supplemented the figure towards extended data: p. 13 – l. 30:

In the extended data (Fig. 2), we illustrate how the polarization imprint can also be simulated for the other cases of Fig. 2 (a)-(c) [main text]. While the first column reproduces the earlier experimental images (for sample detunings of $\Delta\Omega_a=0.47$ deg, $\Delta\Omega_b=1.07$ and $\Delta\Omega_c=1.87$ deg, respectively), the second column juxtaposes the TLS simulations with coupling according to Eq. (M7). For comparison, we also include a simplified version of constant, isotropic coupling in the third column of extended data Fig. 2. Finally, its fourth column visualizes the phase-matching condition for each case. This influences the signal via Eq. (M7), rendering it predominantly sensitive to EUV-quanta that are polarized in the same direction as the reciprocal lattice vector G . As such, all side-lobes of the XPDC cones (i.e., for phasematching $\mathbf{k}_\gamma^{\text{eff}}$ out of plane) show pronounced signal strength, because there is a component of its polarization along G to be imaged. In contrast, the polariton's propagation direction ($\mathbf{k}_\gamma^{\text{eff}}$) at the upper or lower point of the cone may align with G , rendering its polarization largely orthogonal. This results in signal suppression, which is most visible in Fig 2 (b) around $\chi = 0$. Similarly, for the conditions shown in Figs. 2 (a) or (c), $\mathbf{k}_\gamma^{\text{eff}}$ aligns with G at the lower- or uppermost phase-matching point, respectively. Again, this leads to signal suppression - while on each opposite side of the cone, the EUV-idler's polarization is allowed to be parallel to G , giving rise to strong signal and an overall 'horseshoe'-shape.

* What determines the value of V / Γ ? Is there a way to tune their ratio (e.g., via the sample temperature, G vector, idler frequency, ...?)

The reviewer remarks on a very interesting point, namely tuning the ratio of V / Γ . We are actively looking into possibilities to this end. First of all, with regard to Γ , there are two distinct contributions to be considered, i.e., intrinsic broadening due to decay channels of the polaritonic constituents (Γ_e , Γ_γ) and instrumental broadening ($\Gamma_{\text{inst.}}$, see also our answer #2 to reviewer #1).

The latter could be modified by changing, for instance, the transmission bandwidth of monochromator or analyzer, respectively. A systematic study of these influences would require repeated exchange of instrumentation, for which we hope to be granted dedicated synchrotron beamtime in the future. In contrast, the intrinsic values of V and Γ are determined by the material under study, as its electronic structure governs the nonlinear response for the XPDC process. With the material chosen, the idler energy – in terms of addressing specific resonances – could still have a significant influence on the overall strength (and even shape) of the signal.

We have encountered indications to this end in preliminary measurements as well as in theoretical explorations of our TLS model – with further analyses being currently in progress.

As a separate approach, we are also intrigued by the reviewer's idea of utilizing temperature to influence the EUV-polariton. So far, we can only speculate as to what its effect would be. However, we note with caution that any experiment in this direction will need to carefully disentangle circumstantial influences of temperature via the crystal lattice onto the scattering signal (i.e., Debye-Waller-like

effects on XPDC) from direct impact on the EUV-polariton, which is probed by XPDC.

* What are the prospects of generalizing the scheme beyond EUV energies?

We appreciate the reviewer's impetus to generalize the scheme:

On the detection side, generalizations towards higher photon energies should be straight-forward, whereas pushing for lower idler/polariton energies becomes increasingly more challenging. We are investigating this direction (and have done so in the past [23]), however, background effects from linear Bragg diffraction pose a substantial obstacle to be dealt with.

Regarding the polariton itself, its generalization should also be viable beyond EUV-energies – in line with the reviewer's comment that the universality of polaritons 'is not very surprising'. Provided the setup and resolution allow for it, detecting polaritonic XPDC with idler energies outside the EUV would be an enticing prospect. Especially for x-ray-optical PDC, we would expect valuable insights into the valence electronic structure hybridizing with photonic excitations.

* as a technical remark: the subindices "i" on the idler wave vector are different in text and figures, and different to that on ω_{i}

We thank the reviewer for indicating this inconsistency and have corrected the indices throughout the manuscript.

Finally, we want to thank all three reviewers once again for their constructive feedback and hope that we could address their questions and remarks to satisfaction.

References

[12] Sánchez-Barquilla, M., Fernández-Domínguez, A. I., Feist, J., & García-Vidal, F. J. (2022). A theoretical perspective on molecular polaritonics. *ACS photonics*, 9(6), 1830-1841. <https://doi.org/10.1021/acsp Photonics.2c00048>

[14] Mandal, A., Taylor, M. A., Weight, B. M., Koessler, E. R., Li, X., & Huo, P. (2023). Theoretical advances in polariton chemistry and molecular cavity quantum electrodynamics. *Chemical Reviews*, 123(16), 9786-9879. <https://doi.org/10.1021/acs.chemrev.2c00855>

[15] Li, T. E., Cui, B., Subotnik, J. E., & Nitzan, A. (2022). Molecular polaritonics: Chemical dynamics under strong light-matter coupling. *Annual review of physical chemistry*, 73(1), 43-71. <https://doi.org/10.1146/annurev-physchem-090519-042621>

[23] Boemer, C., et al. Towards novel probes for valence charges via X-ray optical wave mixing. *Faraday Discuss.* 228 (2021): 451-469. <https://doi.org/10.1039/D0FD00130A>

[25] Goodrich, J. C., et al. Imaging of X-ray pairs in a spontaneous parametric down-conversion process. *arXiv preprint arXiv:2310.13078* (2023). <https://doi.org/10.48550/arXiv.2310.13078>

- [31] Yoda, Y., Suzuki, T., Zhang, X.W., Hirano, K., Kikuta, S. X-ray parametric scattering by a diamond crystal. *J. Synchrotron Radiat.* 5.3 (1998): 980-982. <https://doi.org/10.1107/S0909049597020232>
- [33] Eisenberger, P., McCall, S. L. X-ray parametric conversion. *Phys. Rev. Lett.* 26.12 (1971): 684. <https://doi.org/10.1103/PhysRevLett.26.684>
- [35] Tamasaku, K., Ishikawa, T. Interference between Compton Scattering and X-Ray Parametric Down-Conversion. *Phys. Rev. Lett.*, 98, 244801 (2007) <https://doi.org/10.1103/PhysRevLett.98.244801>
- [37] Schülke, W. *Electron Dynamics by Inelastic X-Ray Scattering*. Oxford University Press (2007)
- [38] Krebs, D. *A theory for x-ray-optical wavemixing with applications in spectroscopy and nonlinear crystallography*. Diss. Staats-und Universitätsbibliothek Hamburg Carl von Ossietzky, (2022).
- [39] Blaha, M., Johnson, A., Rauschenbeutel, A., & Volz, J. (2022). Beyond the Tavis-Cummings model: Revisiting cavity QED with ensembles of quantum emitters. *Physical Review A*, 105(1), 013719.
- [40] Törmä, P., and Barnes, W. L. Strong coupling between surface plasmon polaritons and emitters: a review. *Reports on Progress in Physics* 78.1 (2014): 013901. <https://doi.org/10.1088/0034-4885/78/1/013901>
- [50] Santra, R. (2008). *Concepts in x-ray physics*. *Journal of Physics B: Atomic, Molecular and Optical*
- [C1] Silvana Botti and Matteo Gatti, *Fundamentals of Time-Dependent Density Functional Theory*, Chapter 3: The Microscopic Description of a Macroscopic Experiment, pages 29–50. Springer Berlin Heidelberg, Berlin, Heidelberg, 2012
- [C2] S. Kokott et al., Efficient all-electron hybrid density functionals for atomistic simulations beyond 10,000 atoms, *J. Chem. Phys.* 161, 024112 (2024), <https://doi.org/10.1063/5.0208103>